# The actin cytoskeleton plays multiple roles in structural colour formation in butterfly wing scales

Victoria J. Lloyd[1] ✉, Stephanie L. Burg [2], Jana Harizanova[3,4], Esther Garcia[3], Olivia Hill[2], Juan Enciso-Romero [1,5], Rory L. Cooper [1,6], Silja Flenner[7], Elena Longo [8], Imke Greving[7], Nicola J. Nadeau [1,9] ✉ & Andrew J. Parnell [2,9] ✉

Vivid structural colours in butterflies are caused by photonic nanostructures scattering light. Structural colours evolved for numerous biological signalling functions and have important technological applications. Optically, such structures are well understood, however insight into their development in vivo remains scarce. We show that actin is intimately involved in structural colour formation in butterfly wing scales. Using comparisons between iridescent (structurally coloured) and non-iridescent scales in adult and developing *H. sara*, we show that iridescent scales have more densely packed actin bundles leading to an increased density of reflective ridges. Super-resolution microscopy across three distantly related butterfly species reveals that actin is repeatedly re-arranged during scale development and crucially when the optical nanostructures are forming. Furthermore, actin perturbation experiments at these later developmental stages resulted in near total loss of structural colour in *H. sara*. Overall, this shows that actin plays a vital and direct templating role during structural colour formation in butterfly scales, providing ridge patterning mechanisms that are likely universal across lepidoptera.

Structural colour produced by the interaction of light with nanostructures enable a diverse and tremendously vivid array of colours[1–4]. They are particularly important in low light environments, for example in the forest understory, as they achieve superior visual signal propagation over pigmentary colour[5]. Despite the importance of biological photonic nanostructures from an evolutionary perspective and as designs for advanced optical materials[6–8], their structural formation remains poorly understood.

Photonic nanostructures within the wing scales are responsible for the structural colour seen in butterflies and moths[9] these include; photonic crystals[10,11], multilayer (Bragg) reflectors[12] and thin-films[13,14]. Each wing scale develops from a single cell, forming a chitinous envelope with an undifferentiated lower layer and a complex structured upper layer covered in longitudinal parallel ridges[15–17]. In numerous structurally-coloured butterfly species, these ridges are composed of multiple layers (lamellae), giving rise to constructive

[1]Ecology and Evolutionary Biology, School of Biosciences, University of Sheffield, Alfred Denny Building, Western bank, Sheffield S10 2TN, UK. [2]Department of Physics and Astronomy, University of Sheffield, Hicks Building, Hounsfield Road, Sheffield S3 7RH, UK. [3]Central Laser Facility—Science & Technology Facility Council, The Research Complex at Harwell, Rutherford Appleton Laboratory, Harwell Campus, Didcot, Oxfordshire OX11 0FA, UK. [4]Core Facility for Integrated Microscopy, Department of Biomedical Sciences, University of Copenhagen, 2200N Copenhagen, Denmark. [5]Department of Biological Sciences, University of Lethbridge, 4401 University Drive, Lethbridge, AB T1K 3M4, Canada. [6]Department of Genetics and Evolution, University of Geneva, Sciences III, Geneva 1205, Switzerland. [7]Helmholtz-Zentrum Hereon, Max-Planck-Strasse 1, 21502 Geesthacht, Germany. [8]Elettra-Sincrotrone Trieste S.C.p.A., 34149 Basovizza, Trieste, Italy. [9]These authors contributed equally: Nicola J. Nadeau, Andrew J. Parnell. ✉e-mail: v.lloyd@sheffield.ac.uk; n.nadeau@sheffield.ac.uk; a.j.parnell@sheffield.ac.uk

interference[18–21]. Ghiradella[19] postulated that developing ridges buckle due to intracellular stress, and that this is responsible for the formation of layered lamellae. These lamellae act as multilayer optical reflector structures which are widely distributed and numerous across butterfly species and this structure has the flexibility to produce colours that span the optical spectrum, from the UV through to the visible spectrum. Inner lumen structures tend to be more optically and structurally complex and there are still many open questions as to how these structures form. Several studies have suggested that these may be patterned by an internal membrane structures formed by the smooth endoplasmic reticulum[11,17], but measurements on adults scales of *Thecla opisena* suggested that cuticle extrusion and folding must be simultaneous processes[22]. However, there have yet to be any direct measurements on developing scales to confirm these hypotheses and in-situ experiments of the developing inner lumen structures are needed to confirm this definitively. In addition, recently F-actin bundles have also been implicated in the formation of elaborate honeycomb nanostructures, specific to Papilionidae[23].

Studying the actin cytoskeleton during scale formation may improve our understanding of how layered lamellae form, as for many cell types actin plays an important role in controlling cell shape[24]. The scale ridges (on which the layered lamellae form) are the result of chitin deposition between parallel actin bundles[16,25,26]. The actin bundles are temporary and stabilized through polymerization and cross-linking of F-actin within developing scale cells[27–29].

The actin cytoskeleton in *Drosophila* bristles, a homologous structure to butterfly scales, has been extensively studied[30]. Genetic knockouts of actin organization proteins have shown the actin cytoskeleton is important in controlling the number and shape of ridges in bristles, as well as localization of chitin synthetase enzymes, required to deposit the ridges[28,31–33]. In butterfly scales, the actin bundles may not just be limited to guiding ridge positioning but could be crucial in sculpting finer-scale aspects of scale morphology, including the photonic nanostructures.

*H. sara*, along with several closely related species in the same clade, are fairly unusual in the *Heliconius* genus in displaying iridescent blue wing colouration[34–36] (Fig. 1A, B). *H. sara* has both structurally coloured blue iridescent and non-structurally coloured black scales (Fig. 1A–C), facilitating direct comparisons of these scale types throughout their development. The structural colour of *H. sara* is generated through layered lamellae on the parallel scale ridges (Fig. 1F, G)[34,35].

Here, we examine F-actin organization during wing scale development in the butterfly *H. sara*, focusing on the formation of the nanostructures responsible for iridescence. Using scanning electron microscopy (SEM) and fluorescence microscopy we investigate whether patterning of F-actin differs between iridescent and non-iridescent wing scales. We use lifetime separation stimulated emission depletion (TauSTED) super-resolution microscopy[37] to gain insight into actin remodelling during scale development. We then chemically perturb the actin dynamics to elucidate whether the actin cytoskeleton plays a direct role in the formation of optical nanostructures in *H. sara*.

## Results

To compare the adult morphology of iridescent and non-iridescent scales on the dorsal forewing of *H. sara* we obtained 3D scans of

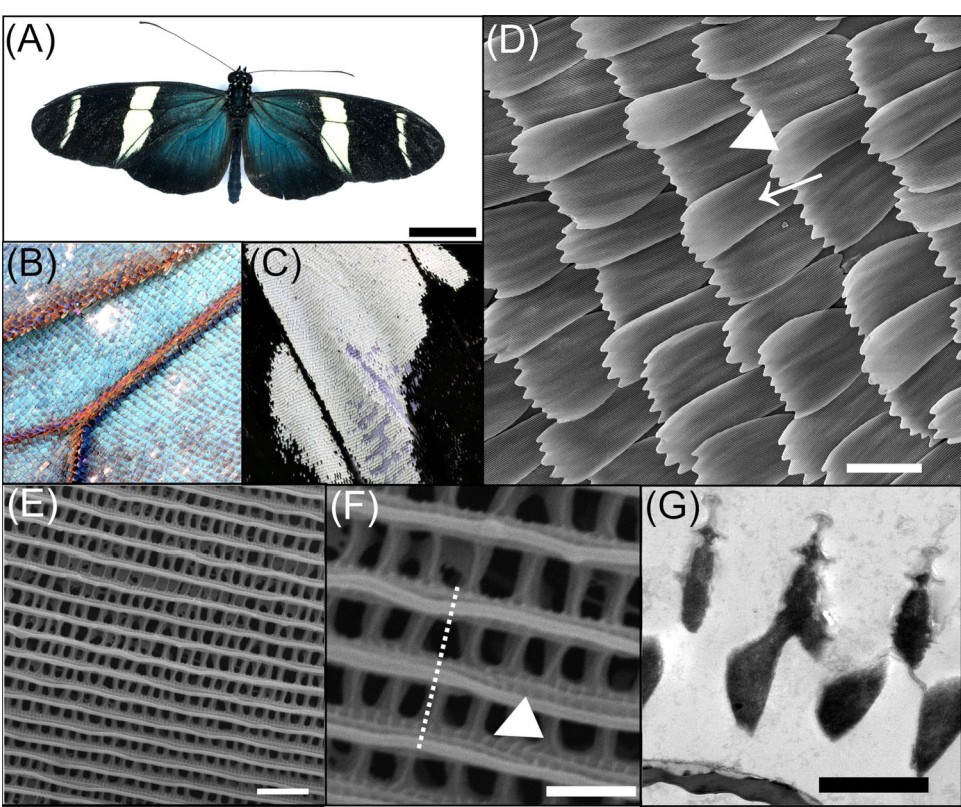

**Fig. 1 | The neotropical butterfly *Heliconius sara*. A** Dorsal view of a *Heliconius sara* individual. **B** Region of blue, iridescent wing scales on the proximal forewing. **C** Region of black and white, non-iridescent wing scales on the distal forewing. **D** SEM image of the overlapping scales on the dorsal wing surface. Cover scales (arrow) sit directly on top of the basal ground scales (arrowhead). **E** Dorsal view of an iridescent wing scale surface, with many periodically ordered longitudinal ridges running parallel to scale length. **F** High-magnification view of an iridescent wing scale showing ridge ultrastructure; with open windows into the scale lumen separated by crossribs. Microribs (arrowhead) pattern the sides of the ridges and are perpendicular to ridge direction (dashed line). **G** Transmission electron microscope (TEM) cross-section through the scale ridges. The layers on the ridges form a multilayer photonic nanostructure. Scale bars lengths: (**A**) 10 mm, (**D**) 50 μm, (**E**) 2 μm, (**F**, **G**) 1 μm.

whole, unmanipulated scales were measured using X-ray tomography (Supplementary Fig. 1, Supplementary Fig. 2 Movie, Supplementary Fig. 3 Movie). The general structure of iridescent and non-iridescent scales is almost identical (S2, S3), with both having a flat smooth lower layer (lamina) and a highly intricate upper layer (lamina). The parallel ridges on the upper lamina are joined together by crossribs, with the spaces between crossribs forming a regular series of windows into the interior scale lumen (Fig. 2C). There was no difference between scale types in their internal structure as measured via X-ray tomography.

Correlation function analysis of the X-ray nano-tomography measured scales indicates a greater crossrib spacing in the black scale compared to the iridescent scale (iridescent 0.483 μm; non-iridescent 0.607 μm)(Supplementary Fig. 1)[38]. An expanded crossrib spacing in black scales likely allows more light to enter the scale and so be absorbed by melanin pigments[39].

To quantify differences in scale morphology we analysed SEM images from 400 scales from the iridescent and non-iridescent wing regions of 20 individuals. Both cover and ground iridescent scales were smaller in size than non-iridescent scales (mean ± SE scale area, blue: cover 2700 μm$^2$ ± 21, ground 3708 μm$^2$ ± 35; black: cover 3044 ± 25 μm$^2$, ground 4123 μm$^2$ ± 30; likelihood ratio, $\chi^2 = 208$, d.f. = 1, $p < 0.001$), which can be attributed to the decreased width of iridescent scales (mean ± SE scale width, blue: cover 29.5 μm ± 0.22, ground 42.9 μm ± 0.3, black: cover 31.3 μm ± 0.22, ground 47.8 μm ± 0.33; likelihood ratio, $\chi^2 = 24$, d.f. = 1, $p < 0.001$; Fig. 2D, S5B).

Having confirmed that the general structure of iridescent and non-iridescent scales are similar, we next quantified differences in the finer scale elements, focusing first on the parallel ridges (Fig. 2C, S4). The iridescent blue scales had significantly reduced ridge spacing compared to the non-iridescent black scales (mean ± SE ridge spacing, blue 0.804 μm ± 0.007, black 0.962 μm ± 0.004; likelihood ratio, $\chi^2 = 446$, d.f. = 2, $p < 0.001$; Fig. 2E, F) confirming the tomography data (Supplementary Fig. 1E, F). This within-wing difference is consistent with prior work comparing between species and populations, which found iridescent *Heliconius* species have reduced ridge spacing compared to non-iridescent species (S6)[34]. The decreased ridge spacing in iridescent scales can be attributed to an overall increase in ridge number, rather than a smaller scale width, with iridescent scales consistently having a greater ridge number for a given scale width (Fig. 2D). There was also an effect of scale type (cover or ground) on ridge spacing (likelihood ratio, $\chi^2 = 27$, d.f. = 2, $p < 0.001$). Iridescent cover scales had significantly reduced ridge spacing compared to iridescent ground scales (Tukey comparison, $p < 0.001$), but there was no difference in ridge spacing between cover and ground scales for non-iridescent scales (Tukey comparison, $p = 0.633$).

Ridge width was slightly greater in iridescent scales compared to non-iridescent scales (mean ± SE ridge width, iridescent 0.315 μm ± 0.002, non-iridescent 0.302 μm ± 0.001; likelihood ratio, $\chi^2 = 43$, d.f. = 1, $p < 0.001$; Fig. 2F). There was no difference in ridge width between cover and ground scales (likelihood ratio, $\chi^2 = 0.24$, d.f. = 1, $p = 0.622$, Fig. 2F). Interestingly, the distribution of ridge width in iridescent scales was much greater than that of non-iridescent scales (S5C).

In general, the morphology of adult *H. sara* iridescent and non-iridescent scales is similar, however there are distinct differences in respect of the ridges. Iridescent scales display slightly thicker ridges and reduced ridge spacing compared to non-iridescent scales. We did not quantify differences in the layering of lamellae within the ridges, which is responsible for the iridescent colour[34,35], as this was beyond the resolution of the x-ray nano-tomography and would have required TEM sections of the ridges. The increase in ridge density and the number of ridges contributes to the increased reflectance of the iridescent scales[34,40].

## Development of *H. sara* scales
We characterized scale development from 25% to 62.5% of pupation, this encompasses scales emerging from the wing epithelium to formation of the final scale morphology (Fig. 3). Overall, the development of iridescent and non-iridescent scales was very similar, and comparable to that reported for other butterfly species[26]. At ~25%, nascent scales begin to emerge, as small actin-dense cytoplasmic projections from the wing epithelium (Fig. 3A). Scale cell nuclei sit directly within the wing epithelium and are considerably larger than surrounding nuclei. Alpha-tubulin staining at 31% reveals the emerging scale buds are rapidly filling with cytoplasm (Fig. 3B, C) and beginning to differentiate into cover and ground scales, with the larger ground scales containing more cytoplasm. In some cases, the tubulin appears organized into dense arrays, suggesting ordered microtubules are beginning to form (Fig. 3C). By 37.5% the scales are essentially elongated sacs, containing thick longitudinal actin bundles (Fig. 3D). Previous research has shown that actin bundles are required for scale elongation. These form through polymerization of actin into filaments (F-actin), followed by cross-linking of filaments together into thick bundles[25,26]. The actin bundles are most clearly discernible at the proximal portion of the scale where it buds from the epithelium through the developing socket (Fig. 3F).

At 50%-56% the scales become flattened and long finger-like projections form on the distal tip (Fig. 3G–I). At this stage, the actin bundles are highly ordered in appearance and cover the entire proximal-distal portion of the scale (Fig. 3I). At around 62.5–69% chitin is deposited between the parallel actin bundles to form the cuticle ridges.

## F-actin patterning differs between developing iridescent and non-iridescent scales
We determined the optimal developmental stage to quantify actin organization as 50% of total pupal development. At this stage, actin bundles are highly regular and have reached the distal portion of the scale (Figs. 2I, 3I)[25,26]. Additionally, chitin ridge deposition is beginning, suggesting that the actin bundles are correctly positioned for ridge formation to occur.

We quantified the spacing and thickness of actin bundles within developing scales using confocal microscopy of phalloidin-stained wings ($n = 12$) (Fig. 2G–L). Iridescent scales had slightly thinner actin bundles compared to non-iridescent scales (mean ± SE bundle width, iridescent 0.438 μm ± 0.004, non-iridescent 0.456 μm ± 0.003; likelihood ratio, $\chi^2 = 19$, $p < 0.001$; Fig. 2J), although this may be influenced by slight differences in development stages observed between the proximal and distal forewing scales[26]. The developing iridescent scales had reduced actin spacing compared to the non-iridescent, black scales (mean ± SE bundle spacing, iridescent 1.07 μm ± 0.02, non-iridescent 1.22 μm ± 0.03; likelihood ratio, $\chi^2 = 40$, $p < 0.001$; Fig. 2K). Furthermore, we found that iridescent scales had a greater number of actin bundles compared to non-iridescent scales (mean ± SE actin bundle number, iridescent 40 ± 1.8, non-iridescent 32 ± 1.1; likelihood ratio, $\chi^2 = 11$, $p < 0.001$; Fig. 2L).

This result is consistent with previous findings[16,26], indicating a tight coupling between the spacing of actin bundles and spacing of chitin ridges, for both iridescent and non-iridescent scales. The mean number of actin bundles in iridescent and non-iridescent cover scales closely matched the mean number of ridges measured in adult cover scales of both types (Fig. 2L). Our results show that the patterning of actin in developing *Heliconius* scale cells plays an important role in governing the density of adult scale ridges, which is an important morphological parameter controlling the iridescent properties.

## TauSTED super-resolution microscopy reveals detailed remodelling of the actin cytoskeleton
To investigate the ultrastructural remodelling of the actin cytoskeleton during the development of *H. sara* scales we used TauSTED

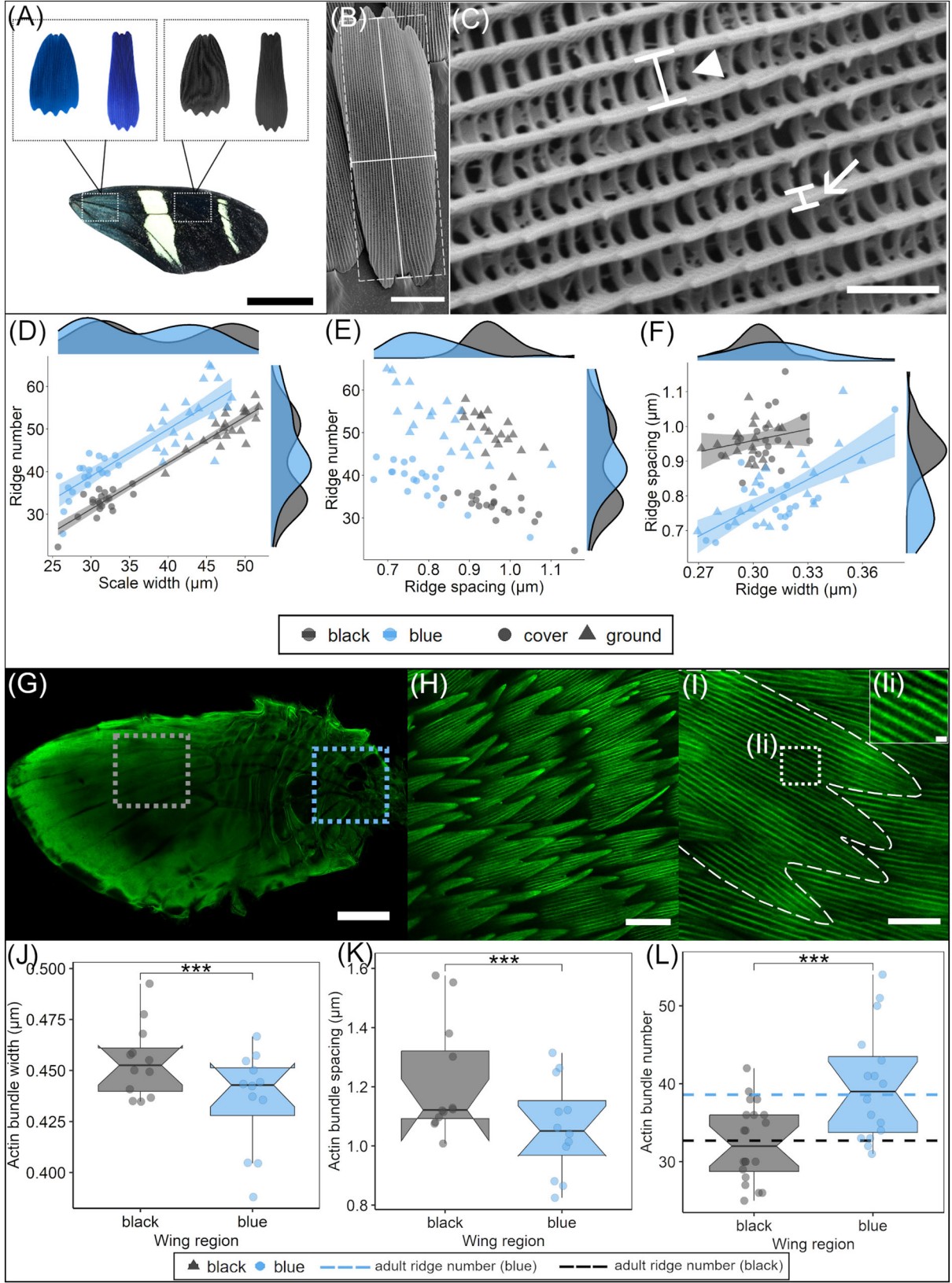

microscopy (Figs. 4, 5, S8). At 44% of pupal development we observe both smaller peripheral actin bundles as well as larger internal actin bundles, described previously by Dinwiddie et al.[26], (Fig. 4A–C). In scales with incipient finger formation, these were seen as vertices forming on the previously smooth distal edge, giving the tip of the scale a trapezoid-like shape. The finger origins coincide with the locations at which prominent internal actin bundles appear to attach to the distal membrane (Fig. 4C, arrowheads; S7, animation). This hints at a possible role of these larger internal actin bundles in specifying spatial positioning of the fingers. Previous actin

**Fig. 2 | Morphological analyses of adult ridge organization and pupal actin patterning. A** Cover and ground scales (SEM images, shown in false colour) were sampled from the proximal, iridescent (blue) wing region and the distal, non-iridescent (black) wing region. Representative SEMs showing measurements of (**B**) scale length (vertical solid line), width (horizontal solid line) and approximate area (dashed line); and (**C**) ridge spacing (arrowhead) and ridge width (arrow). Comparison of cover and ground scales in blue and black wing regions for (**D**) ridge number and scale width (μm) (**E**) number and ridge spacing (μm) (**F**) ridge spacing (μm) and ridge width (μm). Each point is the mean value grouped by individual, region, and scale type. Shaded areas around regression lines indicate 95 % confidence intervals. Density plots on the axes give the distribution of each parameter for iridescent and non-iridescent scales separately (cover and ground combined). **G** whole-mounted phalloidin-stained *H. sara* forewing, showing the iridescent region (blue box) and non-iridescent region (grey box). **H** overlapping wing scales at 50%, with actin bundles visualized through phalloidin staining. **I** Extraction of measurements of actin bundles from an individual developing scale. **Ii** High-magnification zoom of the individual actin bundles showing the spacing between two adjacent bundles. **J** Actin bundle width (μm) for 50% iridescent (blue) and non-iridescent (black) scales. **K** Actin bundle spacing (μm) for 50% iridescent (blue) and non-iridescent (black) scales. **L** Actin bundle number for iridescent (blue) and non-iridescent (black) scales, dashed lines indicate ridge number in adult cover scales. Points in (**J, K**) represent mean measurements for each individual ($n = 12$) grouped by region, points in (**L**) represent individual scales (black $n = 20$, blue $n = 16$). For the box and whisker plots in **J, K** and **L** the box represents the interquartile range (IQR), the horizontal line the median, and the whiskers the extent of the data up to 1.5xIQR. Also *** indicates a significance (*p*) value of <0.001 (likelihood ratio test). Scale bar lengths: (**A**) = 10 mm, (**B, H**) = 20 μm, (**C**) = 2 μm, (**G**) = 1 mm, (**I**) = 10 μm, (**Ii**) = 1 μm.

inhibition experiments performed by Dinwiddie et al.[26], resulted in scales lacking fingers, consistent with a role for actin bundles in specifying finger position and elongation.

At 50% the actin bundles are maximally spaced in agreement with our confocal microscopy observations (Fig. 3). Z-stacks of the optical sections suggest re-structuring of the actin bundles, with the continuous uniform actin bundles, now displaying a more intricate ultrastructural arrangement (Fig. 4D–F). In addition, some actin appears to be present between the large bundles (Fig. 4F), reminiscent of the transient 'actin snarls' described in *Drosophila* bristle development[28,33,41,42].

At 63% of development the large actin bundles are undergoing disassembly, with fracturing of the bundles into disjointed sub-bundles (Fig. 4G–I). A previously undescribed second population of branched actin is now present and is particularly evident at the scale edges as well as at tips of the scale fingers (Fig. 4I). These branched actin filaments are smaller in diameter, located more internally (Fig 4Gi) and are orientated multi-directionally compared to the actin bundles. Along the scale edge, multiple filaments appear to radiate like spokes from single points further inside the scale that connect with the scale edge (Fig. 4I).

At 75-81% the actin cytoskeleton undergoes a final, further reorganization with a highly branched network present in the fingers, radiating towards the distal tips (Figs. 4J–L, 5C). In contrast, the main scale body is now devoid of any parallel actin bundles and is instead entirely filled with square 'blocks' of actin which run the length of the scale and sit between the cuticle ridges. Beyond 81% of development this remaining actin network shows evidence of dissociation (S8 C, F), beginning at the peripheral margins of the cell. This suggests the actin network may be withdrawing from the cell upon completion of cuticle deposition. At 87.5% and beyond TauSTED imaging was not possible due to the presence of pigments.

The remodelling of the actin cytoskeleton throughout the later developmental stages follows the trajectory of cuticle deposition from the ridges to the crossribs (Fig. 5). We also observed potential direct templating roles of actin in distinct ultrastructures including ridge layers, microribs and crossribs. At 63% development we observed a scale finger which was angled in such a way that a side profile of the ridge was visible. Directly below the cuticle ridge layers we noted layers of actin filaments (Fig 5Ai) which apparently matched the layering of the cuticle. At 69% we similarly observed a ridge side-profile which showed many small filaments of actin angled from the vertical along the side of the ridge (Fig B, Bi). This patterning of actin strongly resembles the final positioning of microribs along the ridges (Fig. 1F). Finally, at 75% of development enlarged Z-stacks indicate that square blocks of actin form around the interior of the crossribs, though they do not fill the entirety of the nascent windows (Fig. 5C, Ci).

Utilizing super-resolution microscopy during butterfly scale development, we have revealed new insights into actin cytoskeleton remodelling in butterfly scales. We have shown that the actin cytoskeleton plays a multifaceted role in butterfly scale development, from specifying finger location to a role in the development of ultra-structures, such as the crossribs and windows.

## Actin has universal patterning mechanisms across multiple butterfly species

To confirm the universality of actin patterning mechanisms across butterfly species we again used confocal and TauSTED microscopy to image developing wings scales of *Morpho helenor* and *Parides arcas*.

We chose to study black *P. arcas* scales as these have pronounced ridge structures. A recent study examining developing *P. arcas* green scales using fluorescence imaging saw actin reorganisation similar to our observations in *H. sara*, but around honeycomb structures, which act as optical diffusers, and sit above the gyroid structural colour elements[23]. Other studies[43] have seen wide variation and diversity in the types of photonic structure exhibited in species of *Parides*, including gyroid, lumen multilayer and ridge reflectors. A possible explanation for this variability of inner lumen structures has been suggested by tuning composition, similar to synthetic block copolymer nanostructures, in which the morphology is determined by the control of the constituent block volume fractions[44].

We observed similarity in the patterning and reorganisation of the actin cytoskeleton across all species of butterfly studied (Fig. 6). At 50% of development, large actin bundles are present between the forming cuticle ridges in both *P. arcas* (Fig. 6A–C) and M. *helenor* (Fig. 6G–I). As previously described in *H. sara* and other butterfly species, these large actin bundles play a role in specifying the location of the cuticle ridges[26,45]. During this developmental stage, we also note the presence of additional actin filaments located between the larger actin bundles in the region where cuticle ridge deposition is occurring. This is particularly evident in the scales of *P. arcas* (Fig. 6A–C; Fig. 7, S9); whose ridge layers exhibit noticeably greater width and a more splayed-out configuration compared to *H. sara* and *M. helenor*, where the ridge layers form tightly packed multilayer reflectors. In *P. arcas* these additional actin filaments are positioned directly within the forming ridges, dorsal to the larger actin bundles and exhibit a flared pattern that mirrors the exact arrangement of the cuticle ridge layers (Fig 6Ai; Fig. 7; S9). The co-imaging of chitin and actin together with orthogonal views of the ridges (Fig. 7C, D; S9) confirms that these actin filaments co-localize with the cuticle ridge layers. Furthermore, this is made easier to discern by spatially quantifying the intensity of the actin and chitin in the ridges (Fig. 7E, F). The actin signal is between the peaks of chitin signal that correspond to the sides of the ridge, highlighting the presence of actin directly within the ridge structure itself. Together, these observations provide additional support for the direct templating role F-actin plays in the creation of cuticle ridge layers, which aligns with our explanation for *H. sara* (Fig. 5A, Ai).

At ~60-65% of development in both *P. arcas* (Fig. 6D–F), and *M. helenor* (Fig. 6J–L), the large bundles of actin between the cuticle

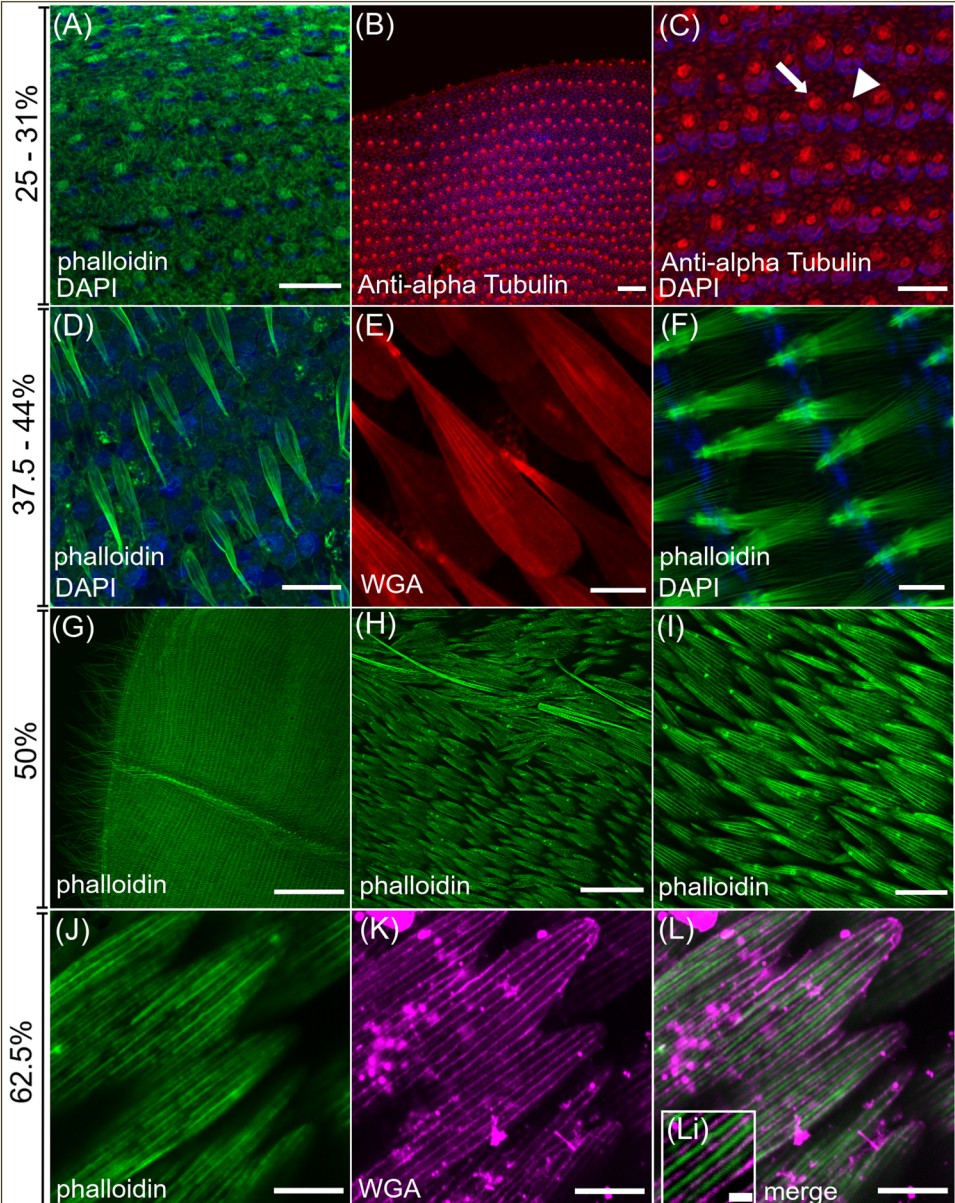

**Fig. 3 | Confocal series of normal wing scale development in *H. sara*.** Cell nuclei counterstained with DAPI (blue). **A–C** Early wing scale development showing cytoplasmic projections from the wing epithelium at 25%. **A** Phalloidin staining (green) of actin in the nascent scales. **B**, **C** Anti-alpha Tubulin immunostaining (red) reveals differing amounts of cytoplasm in developing cover (arrowhead) and ground (arrow) scales and outlines of the socket cells. **D–F** At 37.5–44% the scale cell is a sac filled with organised actin bundles (green) (**D**) and surrounded by a cellular membrane, highlighted by WGA staining (red) (**E**). Forming sockets are clearly visible (**F**) with the actin bundles passing directly through them. At 50–56% (**G–I**) the scales resemble adult scales (Fig. 1D). The distal forewing (**G**) shows hundreds of developing scales. **H** overlapping wing scales adjacent to a wing vein with actin-rich hairs protruding from the vein (arrowhead). **I** The actin (green) within the scales is highly organized at 50% and extends to the proximal portion of the scale fingers. **J–L** final stages of scale development. **J** gaps between the phalloidin stained actin bundles (green) highlights actin sub-bundling (**K**) WGA (magenta) now stains the chitin being deposited extracellularly (**L**) Merge of actin (green) and WGA (magenta) shows the chitin being deposited between the actin bundles (Li). Scale bar lengths: (**A**, **B**, **E**, **I**) 20 μm; (**C**, **D**, **F**, **J**, **K**, **L**) 10 μm; (**G**) 300 μm; (**H**) 50 μm; Li 2 μm.

ridges have dissociated (Fig. 6D–F) and a highly branched network of actin filaments is visible, like *H. sara* wing scales at the equivalent development stage (Fig. 4G–I). In *P. arcas*, we observe a comparable branched actin network to that in *H. sara* scales at a corresponding developmental stage (Fig. 6Di, F; Fig 4Gi, I). This network features an internal arrangement of branched actin filaments extending outward toward the cell edge from singular points located several microns within the scale. In *M. helenor*, the branched network is most discernible in the region between the chitin ridges (Fig. 6J, L), with F-actin regularly interspersed in a perpendicular arrangement between adjacent chitin ridges along the length of the scale, similar to what is seen at a comparable stage in *H. sara*.

Overall, our observations in three phylogenetically distinct butterfly species and different scale types, confirms the universality of a complex and highly dynamic network of actin cytoskeleton in developing scales. In all three species, the actin cytoskeleton displays similarity in its patterning and rearrangement, prefiguring diverse scale ultrastructures throughout scale cell formation, suggesting that patterning is conserved.

### The actin cytoskeleton plays a direct role in optical nanostructure formation

To determine whether actin plays a direct role in optical nanostructure formation, we injected pupae with Cytochalasin D (cyto-D), which

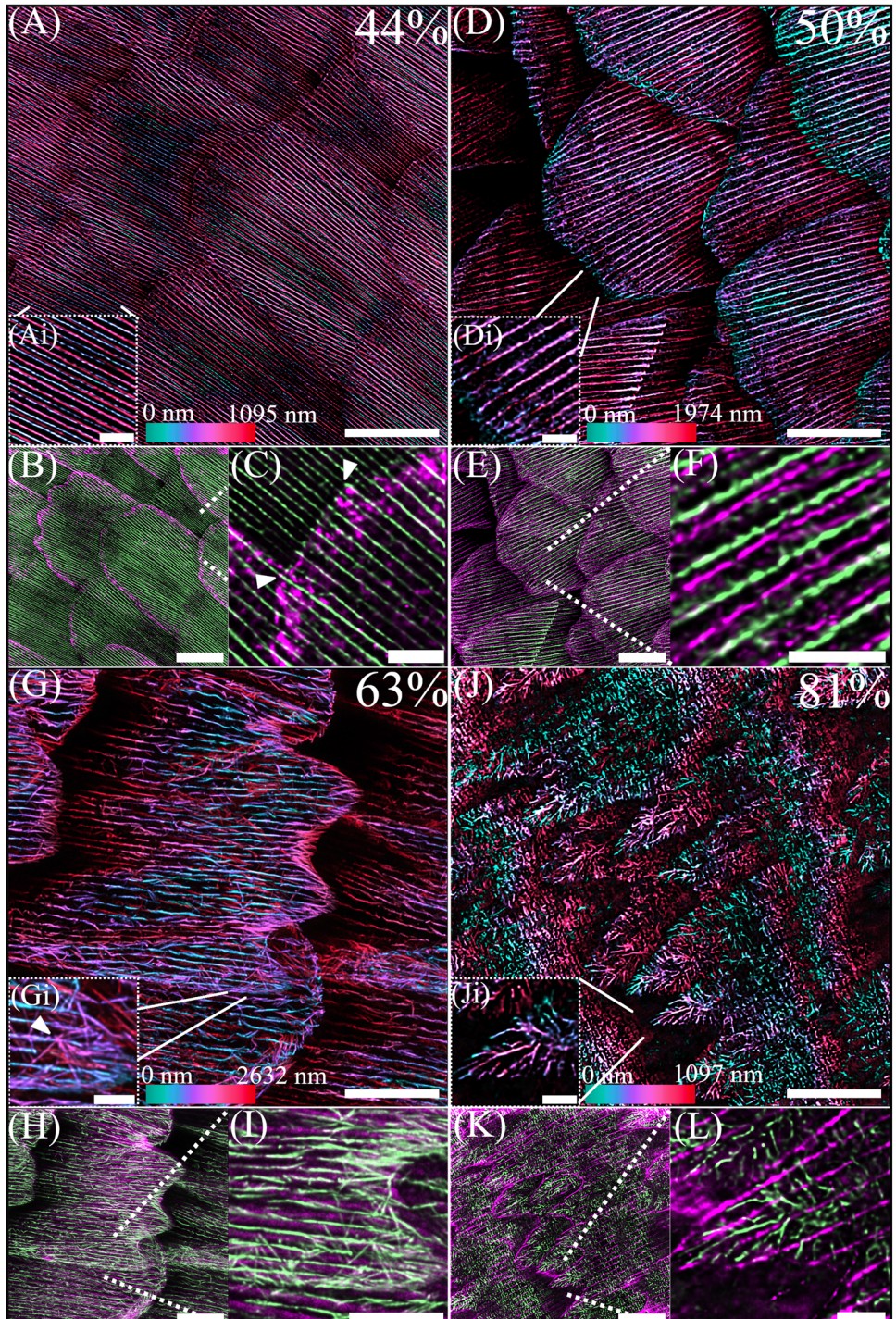

**Fig. 4 | TauSTED super-resolution microscopy of the rearranging actin cytoskeleton during the development of *H. sara* scales.** Depth coloured images (**A**, **D**, **G**, **J**) show F-actin stained with phalloidin. Coloured images below (**B**, **C**, **E**, **F**, **H**, **I**, **K**, **L**) show a merge of actin (phalloidin, green) and chitin (CBD-TMR, magenta). **A**–**C** At 44% of development small, numerous actin bundles are visible which extend to the distal edge of the cell. Colour depth profiles (**Ai**) indicate smaller actin bundles are present on the dorsal surface of the cell (blue) whereas larger actin bundles are located more internally (red/magenta). Incipient cuticle formation begins at the periphery of the scale cells (**B**). Points of finger origination (arrows in **C**) correspond to locations of larger, internal actin bundles associating with the scale tip. **D**–**F** At 50% the actin bundles are maximally spaced as the scale cell becomes increasingly flattened. Cuticle formation is evident across the scale (**E**), with cuticle ridges appearing in between actin filaments (**F**). **G**–**I** At 63% the large continuous, parallel actin bundles are dissociating. A second network of branched F-actin is located more internally (blue in **G**) and is particularly evident along the scale edges and the fingers. Many of the individual filaments appear to radiate from single point of origination (arrow in **Gi**) and span across several microns before apparently attaching to the edge of the scale cell. **J**–**L** At 81% the actin network within the cell undergoes a final rearrangement. At the fingertips, highly branched actin projects from within the middle portion of the fingers towards the distal edges (**Ji**). Within the scale body no parallel bundles or branched filaments are visible, instead the actin has taken on 'block' like appearance. Ridge cuticle formation is complete and ultrastructures such as the crossribs are visible (**K**, **L**). Scale bars: (**I**) 5 μm; (**A**, **B**, **D**, **E**, **G**, **H**, **J**, **K**) 10 μm; (**C**, **F**, **L**, **Ai**, **Di**, **Gi**, **Ji**) 2 μm.

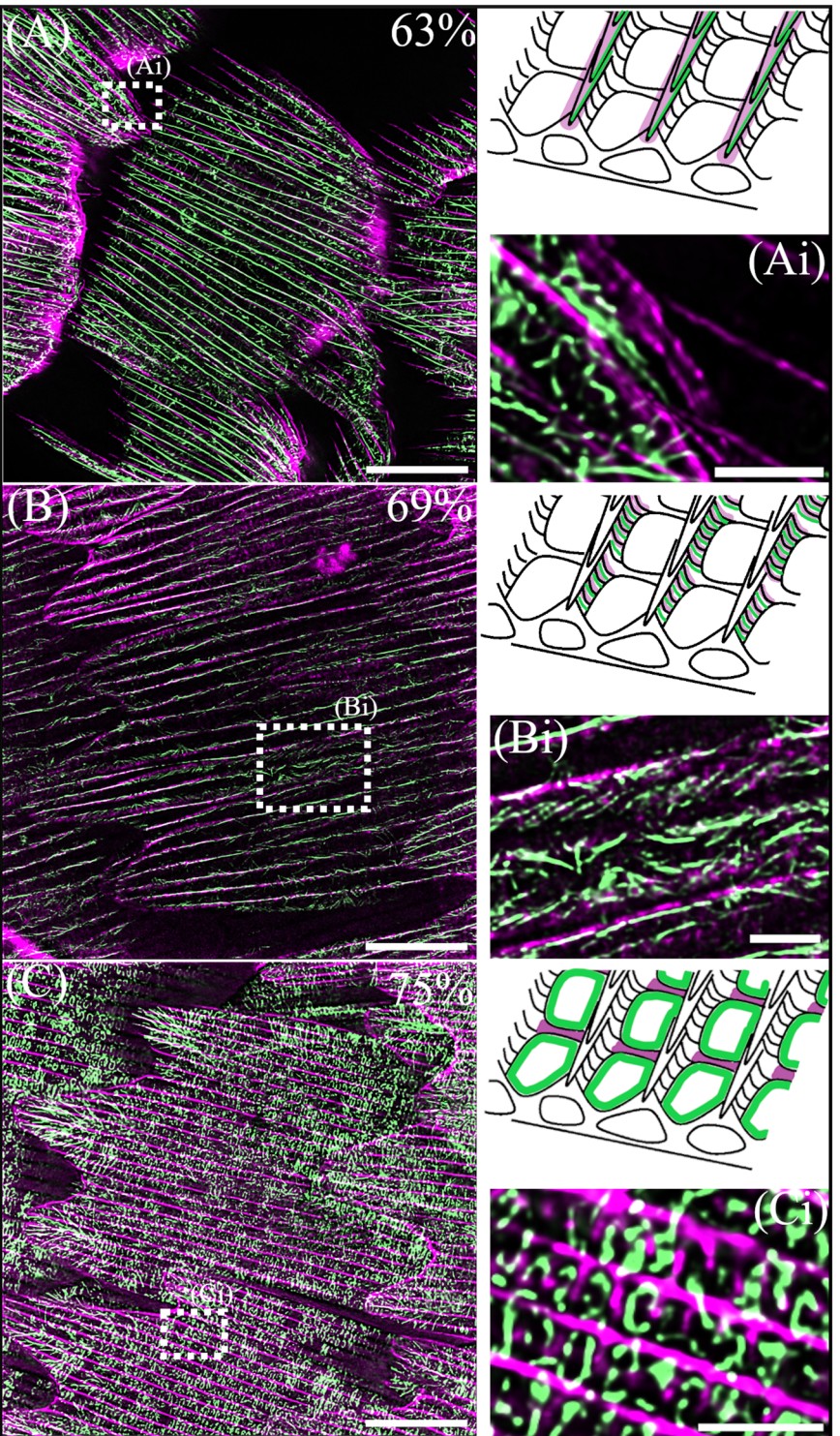

**Fig. 5 | TauSTED super resolution imaging of actin filaments associated with various scale ultrastructures.** Merge of actin (phalloidin, green) and chitin (CBD-TMR, magenta). Cartoon insets highlight the purported location of the actin filaments (green) and the associated cuticle structure (magenta). **A** Black scale at 63% development, with (**Ai**) showing an enlarged view of a scale finger. The ridges at the edge of the scale finger are angled out of plane revealing the layering of a cuticle ridge and actin filaments below. **B** Black scale at 69% development, with numerous individual actin filaments patterned along the side of a ridge (**Bi**), appearing like the microribs of adult scales. **C** Iridescent scale at 75% development. The main scale body is filled with square 'blocks' of actin. The enlarged section (**Ci**) indicates the blocks of actin occur within the window regions in between the crossribs but do not fill these regions entirely. Scale bars: (**A**–**C**) 10 µm; (**Ai**, **Bi**, **Ci**) 2 µm.

inhibits actin polymerization and causes actin bundle disruption[42]. Pupae were injected at 50% development, after ridge spacing is set but before ridge ultrastructures form and during incipient chitin deposition, to assess the effects of actin disruption specifically on structural colour production[26].

We observed substantial loss of structural colour in cyto-D treated forewings, with wings appearing visibly darker in colour (Fig. 8A, B) compared to the non-injected left forewing of treated individuals (Fig. 8A), and the right forewing of controls injected with Grace's Insect Medium (Fig. 8C, D). Reflectance spectroscopy confirmed a significant

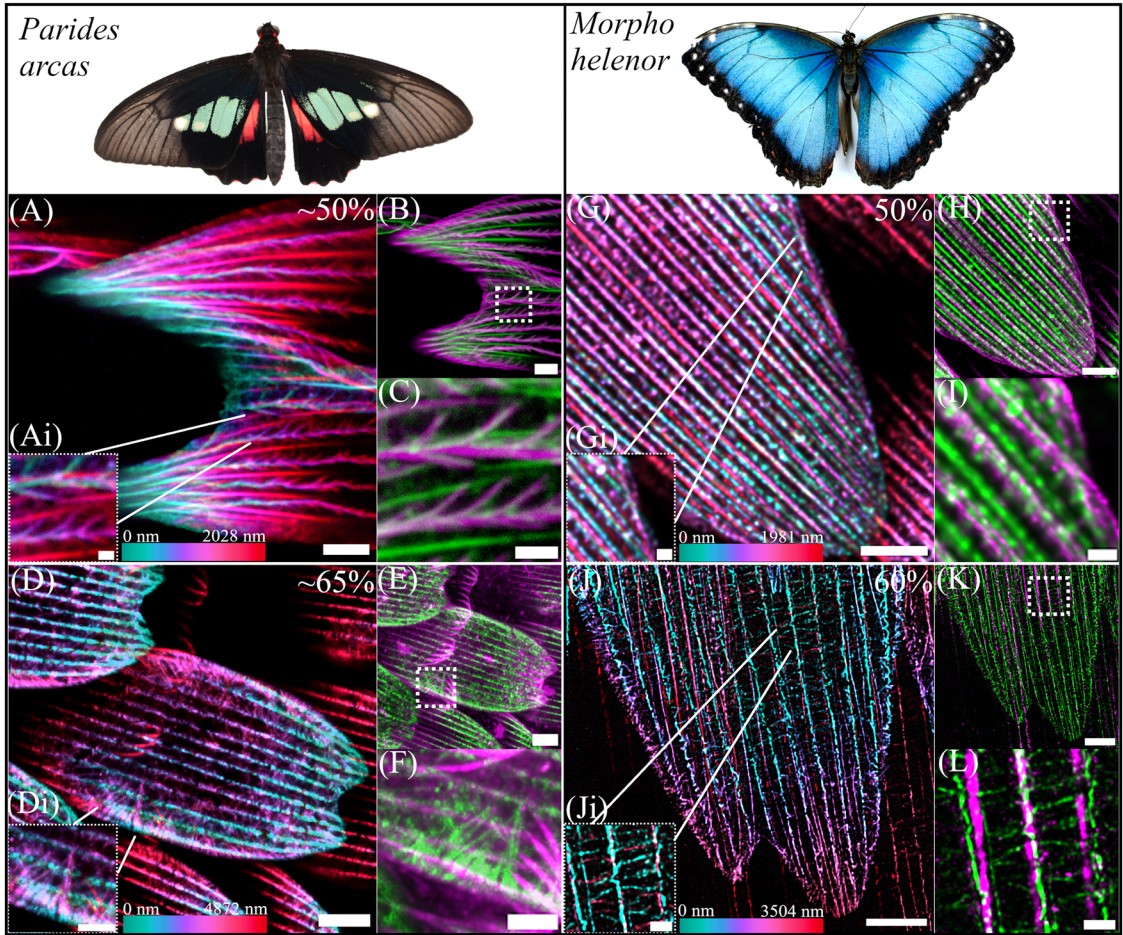

**Fig. 6 | Actin patterning in the developing scales of *Parides arcas* (A-F) and *Morpho helenor* (G-L).** Depth coloured images (**A**, **D**, **G**, **J**) show F-actin stained with phalloidin. Coloured images (**B**, **C**, **E**, **F**, **H**, **I**, **K**, **L**) show a merge of actin (phalloidin, green) and chitin (CBD-TMR/WGA, magenta). Location of enlarged merged images (**C**, **F**, **I**, **L**) shown by dashed boxes in (**B**, **E**, **H**, **K**). **A**–**C** *P. arcas* scales at ~50% of development. Large actin bundles are present between the depositing cuticle ridges in addition to a network of actin within the forming ridges. Colour depth profile (**Ai**) indicates a more ventral positioning of the large actin bundles (red) in comparison with the branched network within the ridges (magenta/blue). Merged images (**B**, **C**) indicate that the actin within the ridges colocalizes with the cuticle ridge layers. **D**–**F** *P. arcas* scales at ~65% development. The large continuous actin bundles have dissociated but some actin remains within the ridges. A second highly branched F-actin network is present at the edges of the scale and is located more internally (magenta in **Gi**) than the actin within the ridges (blue/green in **Gi**). The individual branches of actin originate from single points several microns within the scale and radiate outwards towards the scale edge. **G**–**I** *M. helenor* scale at 50% development. Large actin bundles are present between the depositing cuticle ridges as well as some actin present within the ridge. **J**–**L** *M. helenor* scale at 60% development, the large bundles of actin have dissociated. A branched actin network is now visible, especially between the cuticle ridges (**L**) where individual filaments are positioned at regular intervals perpendicularly to the direction of the ridge. Scale bars: (**A**, **C**, **D**, **E**, **G**, **H**, **J**, **K**) 5 μm; (**Ai**, **Gi**, **I**, **Ji**, **L**) 1 μm;(**D**, **F**) 10 μm; (**B**) 2 μm.

reduction in brightness (*t*-test, *t* = 4.34, d.f. = 33, *p* < 0.001) and flattening of the peak reflectance curve compared to control wings (Fig. 8E). From the individual spectra plots (Supplementary Fig. 10), most cyto-D treated individuals exhibited a completely flat reflectance spectrum with no change in angular intensity (i.e., no iridescence) and so loss of iridescent structural colour (Supplementary Fig. 10). Any remaining reflected colour is bluer than the untreated specimens (bluey green). As such, the perturbation of actin using cyto-D has most likely prevented the multilayer ridges from forming their optimal spacing.

We observed no discernible differences in the size of cyto-D treated scales compared to control scales (Fig. 8F, G, J, K). In some extreme cases we observed deformation of scale shape, with flexing of the fingers outwards and a 'pinching' of the central ridges (Supplementary Fig. 11). There was no difference in the average ridge number between cyto-D treated and control scales (*t*-test, *t* = −0.41, df = 5, *p* = 0.70, Fig. 8O). This confirms that by 50% development, ridge number and position has already been established in scale cells.

SEM imaging of cyto-D treated scales revealed significant deformation of ridge structure compared to controls (Fig. 8I, M). This includes loss in ridge uniformity, evidenced by severe curving and collapse of the ridges (Fig. 8H, I and Supplementary Fig. 11). In terms of alteration of the ridge layering, pivotal for controlling the reflected structural colour wavelength and intensity, this is clearly seen in figure Supplementary Fig. 11 (B, C). With the ridge lamellae having morphology like that seen in a typical non-iridescent scale (see Parnell et al.[34] for more examples of *Heliconius* ridge structures). We noted an increase in 'breakpoints' apparent in ridge layers of cyto-D scales (Supplementary Fig. 12A), again more characteristic of non-iridescent scales, compared to the more continuous ridge layering seen in controls (Supplementary Fig. 12 B). We also observed that in some cyto-D treated individuals, the window regions were entirely filled with cuticle (Supplementary Fig. 11B, E). To quantify ridge disruption, we compared curvature (κ) of the ridges between treated and control scales (Fig. 8N). Cyto-D treated scales had significantly greater average ridge curvature (κ) (μm$^{-1}$) compared to controls (mean ± SE curvature (κ),

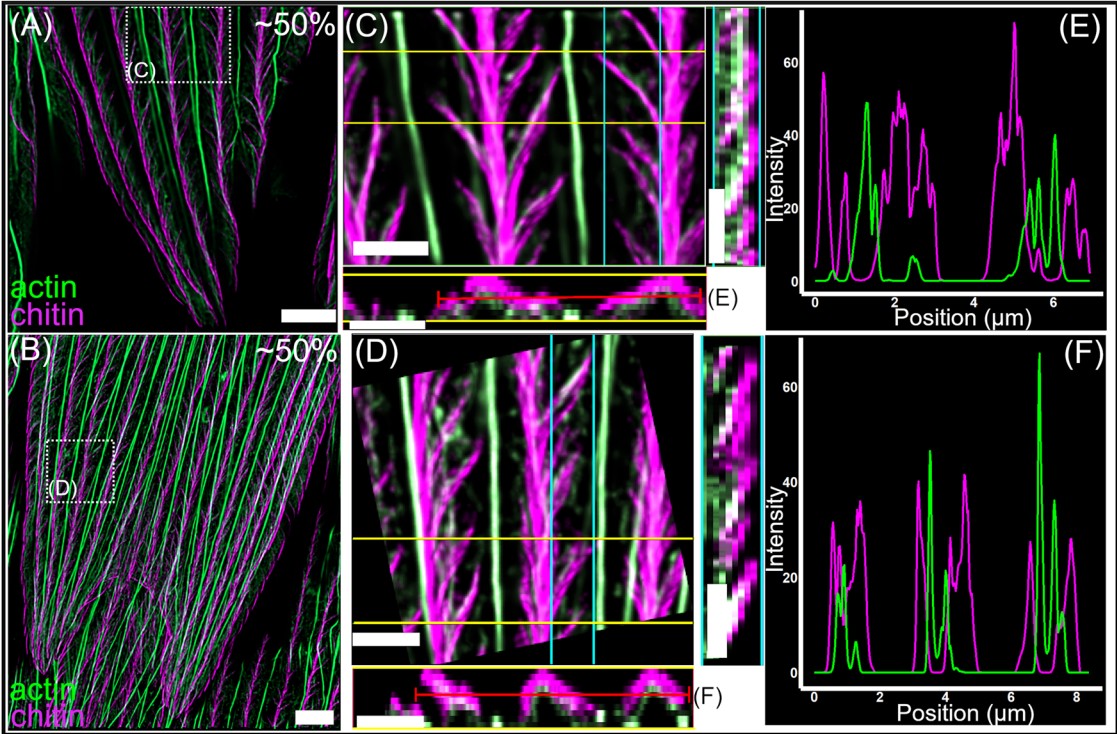

**Fig. 7 | TauSTED fluorescence microscopy of *Parides arcas* wing scales at ~50% development, showing the intimate association of chitin and actin in the developing ridge. A, B** Merge of actin (phalloidin, green) and chitin (CBD, magenta). Location of enlarged merged images (**C, D**) shown by dashed boxes. **C, D** Enlarged images showing a top-down (XY) view of the ridges with the larger actin bundles regularly spaced in between the splayed ridges. Coloured lines indicate the location of the orthogonal views; with the yellow line corresponding to the XZ orthogonal view (below) and the cyan line corresponding to the YZ orthogonal view (right). White pixels indicate colocalization between the actin and the chitin in the orthogonal views. Red line through the ridges in the XZ orthogonal view corresponds to the measured intensities (**E, F**) of actin and chitin by position (µm). Scale bar lengths: (**A, B**) 5 µm; (**C, D**) 2 µm.

treated $0.0566 \pm 0.0018\,\mu m^{-1}$, control $0.0158 \pm 0.0006\,\mu m^{-1}$; *t*-test, $t = -2.78$, df = 12, $p < 0.05$; Fig. 8N). We also noted a large distribution in the average curvature values of treated scales, consistent with the differing levels of scale disruption observed in SEM images.

These results show that perturbation of the actin cytoskeleton during the ridge formation stage results in significant loss of structural colour. This can be directly attributed to damage/disruption of the temporary F-actin scaffold that is used to deposit chitin, registry and coherence in this pre-pattern is vital for producing iridescent colour within the developing ridges.

## Discussion

The gross adult morphology of iridescent and non-iridescent *H. sara* scales does not differ dramatically, showing that only small changes are needed to produce structural colour. This is true for both male and female specimens. However, our results show that iridescent scales have a substantial decrease in the spacing of parallel ridges. Through comparisons between developing iridescent and non-iridescent scales of *H. sara*, we determined that the reduced ridge spacing associated with adult iridescent scales can be attributed to a denser packing of actin bundles during development. Although a relationship between actin bundle spacing and ridge spacing has been shown previously[25,26], and most recently in Papilionidae[23], we show that this association holds for structural colour producing ridges. A tighter ridge spacing is crucial for maximizing reflectance and therefore iridescent scale properties[40,46]. As the layered lamellae responsible for iridescence in *H. sara* are present within these ridges, closer ridge spacing increases the density of light-reflecting surfaces within an individual scale. In the butterfly *Morpho adonis*, (which also contains layered lamellae optical nanostructures), a reduction in ridge spacing of just 0.13 µm yields a 30% increase in reflectivity[46]. Our results show that the actin cytoskeleton is crucial for controlling close spacing of ridges in iridescent scales, through denser packing of actin bundles during the scale development.

The developmental control of total actin bundle number within scale cells warrants further investigation. *Drosophila* bristle studies have highlighted several actin-binding proteins that may be key regulators of actin bundle abundance[28,47]. Perturbation of two such proteins, Actin-binding protein 1 (Abp1) and SCAR, within developing *Drosophila* bristles resulted in extra bristle ridges. These may be promising candidates for future studies of butterfly scale formation[48].

Dinwiddie et al.[26], observed that structurally coloured, silver scales of *Vanessa cardui* possessed double bundles of actin between ridges. In contrast, we observed very little difference in bundle organization between iridescent and non-iridescent scales of *H. sara* (Fig. 2). This is likely due to differences in morphology linked to structural colour production. The iridescent scales of *A. vanillae* have fused windows, with chitin between their ridges, reduced crossribs, and highly patterned microribs. These significant differences in scale architecture are linked to the optical phenomena that *A. vanillae* harnesses to produce structural colour; whose formation involves dramatic shifts in chitin deposition likely controlled by actin patterning[26,31]. In contrast, layered lamellae in iridescent *H. sara* scales are patterned onto an already existing structure – the parallel ridges. There is no dramatic shift in architecture between iridescent and non-iridescent scales in *H. sara* and therefore the actin organization is similar.

If the hypothesis proposed by Ghiradella[19] is correct and F-actin provides the stress forces necessary to induce elastic buckling of the cuticle layer into layered lamellae, then perhaps we should expect to observe differences in actin dynamics, such as compressive forces, rather than large-scale differences in organization. Indeed, our

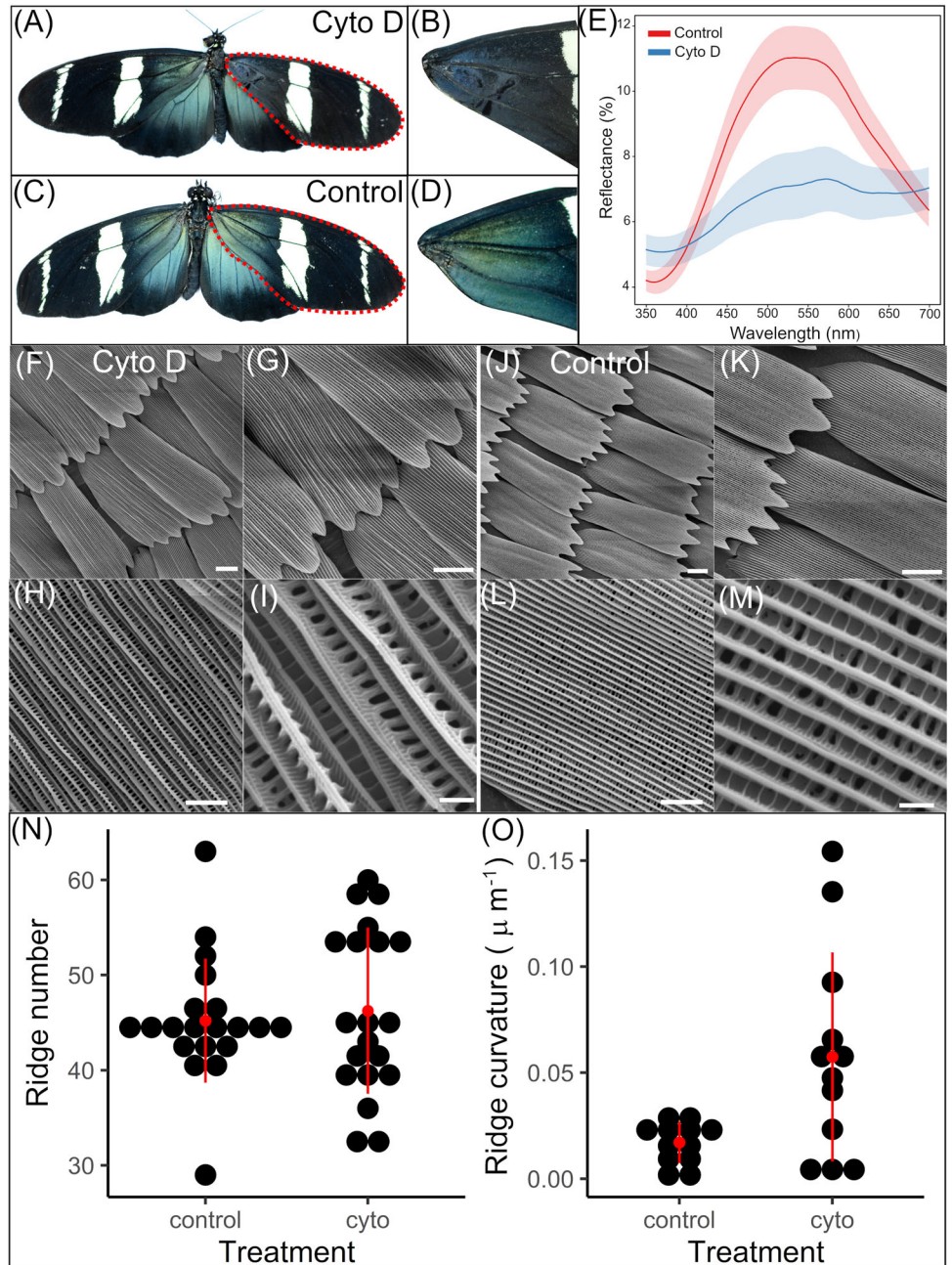

**Fig. 8 | Chemical perturbation of actin with cytochalasin D at 50% development.** Typical morphological phenotype of butterflies injected with cytochalasin D (**A**) and medium/DMSO (control, C) into the right forewing at 50% development and zooms of each (**A → B, C → D**) showing the discernible colour change of the iridescent region. **E** Reflectance spectra at the angle of maximum reflectance for control (red) and cyto-D treated (blue) wings. Shaded areas indicate standard error of the mean (for 21 cyto-D treated and 15 control individuals). **F–M** SEM imaging of typical cyto-d treated (**F–I**) and control (**J, K**) individual's wing scales in the treated region at different magnifications. Differences in brightness of the ridges indicates differences in height. **N** Ridge number for cyto-d treated ($n = 12$) and control scales ($n = 12$). **O** Ridge curvature ($\kappa$) for cyto-d treated ($n = 20$) and control scales ($n = 20$). Black points indicate individual scales. Red points and lines indicate the mean and standard deviation respectively. Scale bars lengths: (**F, G, J, K**) 15 μm, (**H, L**) 5 μm, (**I, M**) 1 μm.

perturbation of the actin cytoskeleton using Cytochalasin D and the resultant dramatic reduction in iridescence (Fig. 8A) support a more direct role of F-actin in controlling the layered lamellae architecture. Cytochalasin D promotes sub-bundling of actin, resulting in wavy and distorted actin bundles within cells[49,50]. We saw that disruption of actin bundles and therefore mechanical integrity during optical nanostructure formation causes considerable reduction in iridescence (Fig. 8).

The deformed ridges observed in our cytochalasin D treated butterfly scales (Fig. 8I) display similarities to bristle phenotypes observed in fly mutants for actin organisation proteins[32,51,52]. As for fly bristles, actin bundles in butterfly scales are crucial for ridge formation, which occurs through extracellular chitin deposition in the inter bundle regions[29,32,41]. Without actin bundles correctly guiding these projections, the final chitin ridges form in an aberrant manner, leading to ridges of varying geometries[32]. We see loss of structural colour in our treated samples, in part attributed to collapse of ridges into varying angles, resulting in the multilayer photonic nanostructures no longer in registry with one another and therefore preventing concerted light reflection.

Interestingly, we observed additional phenotypic effects of actin perturbation on ridge ultrastructure. Harnessing both SEM (Supplementary Fig. 11) and AFM (Supplementary Fig. 12) we noted regular 'breakpoints' appearing on the usually continuous ridge layers. In these images we also see that the lamellae in the ridges have a strong variation in their thickness, this again points to the underlying reason for the loss in photonic properties (Fig. 8I, M). The disruption of ridge layering suggests a further role of actin in directly controlling the formation of layered lamellae. Whether this perturbation of actin disrupts the stress forces needed to buckle the cuticle into layers, as predicted by Ghiradella[19], or instead prevents correct localization of chitin synthase enzymes to deposit the ridges[31] presents an interesting topic for future investigation.

Cytochalasin D may also have disrupted the secondary branched actin network present within scales (Fig. 4I, L). Our TauSTED imaging showed that this network was particularly prominent after 63.5% development, when the chitin ridges had already formed, and the parallel actin bundles were breaking down (Fig. 4G–I). A similar reorganisation of actin was described recently in Papilionidae[23], and here we show that this appears to be universal across the butterflies and involved in the formation of a range of types of optical reflector (they describe its role in forming internal honeycomb lattices, which act as light diffusers and here we find it is also involved in ridge reflector formation). We speculate that this network may be involved in stabilising the scale cuticular structures as the prominent parallel actin bundles break down. During this stage, the scale is still filled with cytoplasm and therefore likely subject to high cytoplasmic pressure[53]. In support of this prediction, we see actin filaments in between the cuticle ridges as well as a high density of branched actin at the scale edges and in the fingers (Figs. 4, 5). Furthermore, some scales treated with cyto-D exhibited loss of overall uniformity, such as splayed fingers, consistent with disruption to a scale-wide stabilising mechanism (S11A). The branched actin filaments may act as a series of intracellular 'struts', keeping the complex cuticular ultrastructure in a fixed registry until cuticle deposition is completed. Interestingly, at 75% development the actin becomes 'block-like' as it arranges around the crossribs (Fig. 5C), suggesting that this stabilizing mechanism of actin may follow the path of the depositing cuticle internally as scale development progresses.

In conclusion, our study shows that the actin cytoskeleton plays a crucial role in the development of structural colour specifically in the formation of ridge reflector nanostructures. Through denser packing of actin bundles during development, iridescent *H. sara* scales attain a higher density of chitin ridges enhancing the optical reflectance. In addition, using actin perturbation experiments, we demonstrate that the actin cytoskeleton likely plays a direct role in the development of layered lamellae. The actin scaffold appears to template chitin deposition across species and may stabilise the chitin structures as they are forming. Absence or diminution of the actin results in photonic structures that are out of registry with one another and causes disruption and lack of continuity in the lamellae that comprise the Bragg reflective layer, leading to substantial changes in the overall reflected intensity and directionality of the structural colour.

We postulate that the role of actin may be akin to the layout and pinning stage used in dressmaking, crucial to achieving high levels of long-range order and perfection across an entire scale cell. This same patterning approach, involving templating by the actin cytoskeleton, is also used by diatoms, a wholly unrelated class of much simpler biological organisms[54]. Therefore, this process could be a conserved (or convergent) pattering route for complex nano- and microstructures across the tree of life. Ultimately, a better understanding of how the actin cytoskeleton controls structural colour development in butterflies will help us understand how such complex natural photonic structures evolved and are patterned within individual cells. This has broader implications for our understanding of intracellular patterning more generally and for the design of synthetic systems to produce photonic materials with similar optical properties.

## Methods

**Butterfly rearing**—Stocks of *Heliconius sara* were established from pupae originally purchased from Stratford-upon-Avon Butterfly Farm, United Kingdom. Adult butterflies were maintained in breeding cages at 25 °C, and fed on 10% sugar water solution with ~1 gram of added pollen per 200 ml. *Passiflora auriculata* was provided for adults to lay eggs on. Caterpillars were kept at 25 °C, 75% humidity and fed on *Passiflora biflora* shoots. Pre-pupation caterpillars were checked regularly, and the time of pupation was recorded as the point of pupal case formation.

A stock of *Morpho helenor* was established with pupae purchased from Stratford-upon-Avon Butterfly Farm, United Kingdom. *Morpho helenor* was raised over several generations in breeding cages at 25°C and adults fed on decaying fruit. *Vicia faba* was provided for egg laying. Caterpillars were kept at 25 °C, 75 % humidity and raised on legume-type plants, and the time of pupation was recorded as the day the pupal case formed.

Wing scale development occurs during the pupal stage[55]. At the desired stage, wings were dissected from pupae in phosphate buffered saline (PBS) and immediately fixed for 15 min in 4% paraformaldehyde in PBS, at room temperature. Developmental stages of pupae were recorded as a percentage of pupal development, with *H. sara* and *M. helenor* taking 8 and 14 days from pupal case formation to eclosion at 25 °C, respectively.

*Parides arcas* pupae were purchased from Stratford-upon-Avon Butterfly Farm, United Kingdom. Recording of the exact pupation times was not possible due to direct purchase from a supplier but these were approximated based on observation of the wing cuticle levels at dissection and the average time of emergence of the remaining pupae. The pupation time for *Parides arcas* can range from 14–21 days[56].

**Electron Microscopy**—Adult wing samples were cut from regions of interest and adhesive tape was lightly applied to remove some cover scales. Samples were sputter coated with gold before being imaged on a JEOL JSM-6010LA SEM, equipped with InTouchScope software. See SI for TEM methods.

**Immunofluorescent microscopy**—Fixed wings were stained with various combinations of: mouse Anti-α-Tubulin primary antibody followed by a Cy3 AffiniPure Donkey Anti-Mouse secondary antibody for microtubules; Phalloidin or SiR-actin for actin; Wheat Germ Agglutin (WGA) for membrane and chitin, which was later replaced by Chitin Binding Domain that is specific for chitin. Slides were stored at 4 °C until imaged. For each slide, both the proximal iridescent region and distal non-iridescent region were imaged. Confocal microscopy imaging was performed on a Nikon A1 confocal laser microscope equipped with NIS elements software. Super resolution imaging was performed on a Leica TCS SP8 STED microscope with Falcon module (see SI for details).

Comparative analyses of iridescent and non-iridescent *scales*—10 males and 10 females, were used for SEM analysis of adult scale morphology with 10 cover and 10 ground scales analysed for each individual. 12 pupae at 50% development were used for phalloidin staining to measure actin bundle number, size and spacing, with 5 scales measured in each wing region (blue vs black). Image analysis was conducted in ImageJ[57] (See SI for details).

**Chemical perturbation of actin** - Actin inhibition experiments followed the protocol of Dinwiddie et al.,[26]. Ready-made cytochalasin D solution (Merck) (5 mg/ml in DMSO) was diluted to a final concentration of 20 μm in Grace's insect medium (Merck). Pupae were injected at 50% pupal development using a Hamilton microliter syringe (701 N). 5 μl of drug was injected directly into the proximal portion of the right wing blade. Control pupae followed the same protocol but were injected with 5 μl of 20 μm DMSO in Grace's insect medium. Pupae were allowed to continue development until eclosion. Immediately after the wings had

dried post-eclosion, butterflies were humanely killed. Butterflies which failed to emerge properly were discarded from further analyses. Only batches with an eclosion rate of over 50% were included in further analyses. A chi-squared test was used to assess differences in emergence rate between control and treated pupae. Whole wing imaging was performed on a Nikon D7000 DSLR camera. Scale imaging was performed using SEM and AFM (see SI).

## Statistical and reproducibility

For the imaging work in terms of sample size we use 'n' to denote an individual wing mounted on a glass slide (an instance), at least one image was taken per wing and often there are multiple scales per image, particularly at the early stages of development. For Figs. 1 and 2A–C, SEM images were taken for 800 scales across 20 individuals, for the TEM image (Fig. 1G) $n = 1$. In Fig. 2G–I $n = 87+$ images from 12 individuals. In the different scale developmental stages in Fig. 3, 25% $n = 2$, 31% $n = 2$, 37.5% $n = 3$, 44% $n = 4$, 50 % $n = 5$ (plus the $n = 12$ from the confocal analysis), 62.5% $n = 3$. For each sample TauSTED images (4, 5, 6 and 7) were taken from both the iridescent blue and non-iridescent black wing scales. Several control wings were imaged during the later development timepoint to check for autofluorescence and/or non-specific binding. For Fig. 4, 44% $n = 2$, 50% $n = 3$, 63% $n = 1$ (+$n = 1$ control), 81 % $n = 2$. Figure 5, 63% $n = 1$, 69% $n = 2$ (+$n = 1$ control), 75% $n = 2$ (+$n = 1$ control). Figure 6 *Parides* 50% $n = 1$, 65% $n = 1$. For the *Morpho* specimen a time series was performed on *Morpho helenor*, with at least $n = 1$ across 6 stages of development. (38%, 43% 50%, 60%, 64%, 71%).

All statistical analyses were performed in R (Version 3.5.2)[58]. For SEM analyses of adult iridescent and non-iridescent *H. sara* scales, we constructed a linear mixed effect model for each response variable (scale area, scale length, scale width, ridge spacing, ridge width) using the lme4 package[59]. Prior to fitting the mixed effect model for ridge width, we averaged individual ridge measurements per scale. For models of ridge spacing, scale area and ridge width we included 'individual' as an intercept only random effect and for the model of ridge spacing, we included an interaction term between scale type and region. For scale length and scale width we fitted a random slope mixed model, allowing a different response to wing region for each individual. We used likelihood ratio tests between models with the Chi-squared distribution to assess statistical significance of sequentially dropped terms. For pairwise comparisons, Tukey multiple comparison tests were performed using the emmeans package in R[60]. For analyses of the ridge spacing between the proximal and distal scales of *H. e. demophoon*, we firstly averaged measurements for each region per individual. Given the lower sample size we performed a paired *t*-test.

For analyses of actin bundle width, bundle spacing and bundle number in developing iridescent and non-iridescent scales, we constructed linear mixed effects models using the Lme4 package. For bundle spacing and bundle width, we firstly averaged bundle measurements per scale to account for multiple bundle measurements. For all models we fitted 'individual' as an intercept only random effect and tested statistical significance using likelihood ratio tests with a Chi-squared distribution. All figures were constructed with ggplot2[61], GIMP (v.2.8.22.)[62] and ImageJ[57]. See Supplementary Information for the R scripts and data used to undertake these analyses.

## Reporting summary

Further information on research design is available in the Nature Portfolio Reporting Summary linked to this article.

## Data availability

The processed imaging data are available at the University of Sheffield ORDA repository (this repository is hosted by Figshare) https://doi.org/10.15131/shef.data.c.7114033. This data is made freely available under a CC BY 4.0. The full image dataset is available from the corresponding author upon request.

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

## Acknowledgements

We thank Dr Alan Dunbar (CBE, University of Sheffield) for use of the scanning electron microscope (funded via the EPSRC grant EP/K001329/1). We are grateful to Chris Hill (Electron Microscopy Facility, University of Sheffield) and Frane Babarović (School of Biosciences, University of Sheffield) for assistance in transmission electron microscopy. Confocal microscopy was conducted at the Wolfson Light Microscope Facility, University of Sheffield, and we thank Darren Robinson for his assistance. We also thank Dr. Natalia Bulgakova and Dr. Miguel Ramirez (School of Biosciences, University of Sheffield) for provision of an Anti-α-tubulin antibody and for their expertise in immunofluorescent staining. We thank William Hentley (School of Biosciences, University of Sheffield) for assistance in the high-magnification imaging of the wings. Finally, we thank Gonzalo Castiella Ona who provided invaluable assistance in the laboratory during his Erasmus funded internship. This research was supported by a Natural Environment Research Council (NERC) Fellowship (NE/K008498/1) to N.J.N. and

doctoral training partnership (Adapting to the Challenges of a Changing Environment, NE/L002450/1) studentship to V.J.L. O.H. was funded by the Biotechnology and Biological Sciences Research Council (National Biofilms Innovation Center) A.J.P. TauSTED imaging was supported by the Science and Technology Facilities Council at the Central Laser Facility (Experiment 20130025). For the purpose of open access, the author has applied a Creative Commons Attribution (CC BY) license to any Author Accepted Manuscript version arising.

## Author contributions

V.J.L., N.J.N., R.L.C. and A.J.P. designed the project. V.J.L. and A.J.P. conducted most of the data collection, with the exception of the X-ray nanotomography (S.F., E.L. and I.G.) and its rendering which S.L.B undertook, and reflectance spectroscopy analysis which J.E.R. performed. J.H. and E.G. designed the STED experiment together with V.J.L., V.J.L. and O.H., acquired, whilst V.J.L. processed the STED data. All authors contributed to data interpretation. V.J.L. wrote the manuscript, with all authors contributing to manuscript revision and editing.

## Competing interests

The authors declare no competing interests.
