## [Peer Review File · Nature Communications]

Reviewers' Comments:

Reviewer #1:

Remarks to the Author:

The manuscript "The actin cytoskeleton plays multiple roles in structural colour formation in butterfly wing scales" by Lloyd and colleagues investigated the role of actin fibers in the formation of nanostructures in butterfly wing scales during their formation. For this, they use state-of-the-art light and electron microscopy as well as STED microscopy to increase their resolution. Focused mainly on *Heliconius* butterflies, with an attempt to show generality by including *Morpho* and *Parides* butterflies, the authors show that actin can reorient and restructure in the scale during formation, after the ridge structures were formed. The work is thorough, the manuscript well written and the figures (mostly) clear.

Most of their impactful conclusion (the re-orientation of f-actin during development) was very recently described by Seah and Saranathan in eLife (<https://elifesciences.org/articles/89082>) for a few different *Parides* butterflies. The findings in this paper make it difficult for me to recommend publication in Nature Communications.

Outside the question of novelty, the manuscript is logically arranged and adds to the growing amount of literature that investigates the in vivo formation of butterfly wing scales, aimed at a specialised audience interested in this question. Some revisions are suggested by this reviewer:

- Introduction: the introduction can be put on a wider context, e.g. by including more review literature on structural colors (thinking of e.g. the McDougal, Schroeder or Isapour reviews), a wider introduction of growth mechanisms (at least mention alternative growth hypotheses for lumen structures), and specify a bit more that ridge multilayers are quite common before you jump into details in lines 72.
- The x-ray tomography dataset seems a bit superfluous (and is not shown in Figure 2, see line 92). Why is it useful and what added data can be extracted from this that couldn't be obtained from SEM images?
- Scale volume: line 116 discusses scale spacing etc. Could this be a volume conservation effect? Why yes/no?
- The differentiation of non/iridescent scales in lines 130-137 needs more work and is hard to follow.
- Chitin and cuticle: these are used synonymously here, yet there is a distinct difference in its biochemical composition and function (see Politi and others). Please rephrase and be careful throughout.
- Errors: many values / times of development can only be seen as approximate as its extrapolated from the mean time to the eclosion of the butterfly. I would at least add a cautious note about the times.
- *Parides* work: The findings of the section starting at line 235 should be compared to ref 29, but also other *Parides* work, e.g. 10.1186/s12862-014-0160-9
- The cyto-D experiments are important and take up most of the discussion of the work. While interesting, the loss of structural color needs to be explained more. Some reflectance is still there and the reflectance curves certainly don't point to "a total loss of structural color" (l. 285).
- Materials: add pupation time for *P. arcas* (not given).

Reviewer #2:

Remarks to the Author:

This paper delves into describing the contribution of the Actin cytoskeleton into the building of intricate butterfly wing scale ultrastructures. Using groundbreaking phenotyping methods to quantifying morphology in the sub-micron range, and super-resolution microscopy (the TauSTED documentation of chitin/actin co-stains are outstanding), the authors radically raise the standard of how chitin modifications and epithelial structures are elaborated. One of the important insights that is brought up is that actin network organization continues to scaffold scale morphology throughout its development, while lower resolution data previously implied depolymerization in the late stages. The authors also complement the *H. sara* data (documenting a proximo-distal shift of melanic scales, from iridescent to non-iridescent), with two comparative models (*Parides* and *Morpho*), which primes interesting future studies (similarly to the recent Seah and Saranathan paper). It is also important to highlight that the article generates a lot of quantifications and

measurements, and made a real effort to analyze these with proper statistics.

While technical in its Results, the article is superbly written and illustrated, with a real effort to bring clarity where it can. I thus expect it to be of interest to a general audience of cell biologists interested in the quirks of actin dynamics as well as other researchers generally curious about intracellular morphogenesis. I highly recommend this article for publication, and think it does not require a significant revision.

Minor comments :

For reproducibility, the methods should further expand the description of the staining protocols, for example detailing the detergent concentration, the incubation times, and the order in which these incubations are made.

In the discussion of Fig.6, I think the article was not particularly explicit about whether the actin data supports a role in building the Bragg's reflector structure itself. Is this because of a staging issue? Since Morpho iridescent scales consist of extraordinary lamellar stacks, I would have expected periodic actin branches pretemplating these extensions. I suspect this will be the subject of further studies, including perhaps in UV-iridescent pierid butterflies described in the Ghiradella foundational papers with TEM, or in other butterflies convergently evolved for ridge stacking (suggested reference : <https://doi.org/10.1242/jeb.245940>)

Reviewer #3:

Remarks to the Author:

In this manuscript, the authors investigated the structural features of butterfly wing scales with a variety of techniques (e.g., SEM, X-rays, TauSTED) and propose that actin plays multiple roles in structural color formation in butterfly wing scales. The experiments were very well executed and the micrographs are very impressive. However, it remains only a descriptive work, and the biology of the relationship between actin and butterfly wing scales remains unclear. I feel that the present form of this paper is not suitable for publication in Nature Communications.

Major concerns

1. Structural color is a well-known research field in nanophotonics and several important results have been published. Structural color is mainly dominated by various features of the cell of nanostructures, such as size, shape, categories of materials and so on. In a way, the scales of butterfly play the similar role as the periodic cells of nanostructure. As literatures (23,25,28) reported that the actin cytoskeleton is important in controlling the number and shape of ridges in bristles. Therefore, I don't doubt the conclusion that actin cytoskeleton plays key roles in structural color formation in butterfly wing scales. In my opinion, the novelty of this work is not sufficient to attract wide attention. Moreover, the authors just propose one potential prospect: providing likely ridge patterning mechanisms. I would encourage the authors to introduce more great and further potential applications for being of interesting to the related scientific communities.
2. This article gave a lot of descriptive work, but I think the evidence to support the important role of actin in the formation of scale-like patterns is insufficient. The only functional experiment was that cytochalasin D could disrupt the pattern of scales, but it also did not rule out the possibility that its effects were due to cell death or other indirect causes. I think it is necessary to observe the influence of cytochalasin D on actin and the pattern of chitin distribution in a short period of time. Authors should also try to stabilize actin with jasplakinolide and observe its effects.

Minor concerns

1. I think there is a difference of Young's modulus between adult and developing *H. sara*. I wonder that if Young's modulus of butterfly wing scales would affect the actin cytoskeleton? Why?
2. Lines 478 mentions that "we constructed a linear mixed effect model for each response variable...", the actin cytoskeleton as a function of scale area, scale length, scale width et al. should be a nonlinear effect. As a result, the data interpretation of this simulated model is not absolutely accurate and sometimes wrong. I recommend the authors to add several descriptions and

explanations on this point.

Thank you again for submitting your manuscript "The actin cytoskeleton plays multiple roles in structural colour formation in butterfly wing scales" to Nature Communications. We have now received reports from 3 reviewers and, on the basis of their comments, we have decided to invite a revision of your work for further consideration in our journal. Your revision should address all the points raised by our reviewers (see their reports below).

REVIEWER COMMENTS

Reviewer #1 (Remarks to the Author):

The manuscript "The actin cytoskeleton plays multiple roles in structural colour formation in butterfly wing scales" by Lloyd and colleagues investigated the role of actin fibres in the formation of nanostructures in butterfly wing scales during their formation. For this, they use state-of-the-art light and electron microscopy as well as STED microscopy to increase their resolution. Focused mainly on *Heliconius* butterflies, with an attempt to show generality by including *Morpho* and *Parides* butterflies, the authors show that actin can reorient and restructure in the scale during formation, after the ridge structures were formed. The work is thorough, the manuscript well written and the figures (mostly) clear.

We thank the reviewer for their assessment of our manuscript, which we think shows generality and universality in the way that butterfly scale structures form, and in particular the way in which ridges structures are formed, a structure widely used for structural colour.

Most of their impactful conclusion (the re-orientation of f-actin during development) was very recently described by Seah and Saranathan in eLife (<https://elifesciences.org/articles/89082>) for a few different *Parides* butterflies. The findings in this paper make it difficult for me to recommend publication in Nature Communications.

Whilst the findings of the recent paper the reviewer mentions are relevant and add to our understanding of scale cell development, the paper by Seah and Saranathan focuses on a specialist honeycomb structure, present only in Papilionidae.

Our work however shows that all the components of the scale cells are pre-templated / patterned by the appropriate and highly regulated placement of actin. This begins with the spacing of the ridges, which are highly correlated at 50% development with the final fully developed ridge scales; this is of direct relevance to the structural colours produced, as ridges are the major way in which butterflies produce iridescence. Therefore, our work both shows that the rearrangement of actin during development is a universal feature of butterfly scale development (expanding from just Papilionidae) and that it is intimately involved in structural colour formation.

Figure R1. The window structure features present in nascent butterfly scales, showing the larger “blocky” actin templated features, of which smaller ones are seen in Papilionidae.

The reorganisation of the actin from parallel fibres to form the inter-ridge “square blocks” of actin is reminiscent of the specialist structures seen by Seah and Saranathan. But we think that the more general structure is the larger window that we have imaged, as such this has importance for many others working in the area of understanding scale structuring and development.

Also it is worth highlighting that all butterflies, as well moths and ancient lepidoptera have a similar generic scale morphology (Ghiradella 2009), with the presence of ridges. The three different species we have studied, two of which are iridescent (Morpho and Heliconius) and one non-iridescent (Parides arcas), all three utilise the same universal actin templating to form the ridge structures. All in all, this shows the generality and universality of the way in which actin patterns these widely conserved scale structures.

Figure R2 (Figure 7 revised manuscript). TauSTED fluorescence microscopy of *Parides arcas* wing scales at ~50% development, showing the intimate association of chitin and actin in the developing ridge. (A, B) Merge of actin (phalloidin, green) and chitin (CBD, magenta). Location of enlarged merged images (C, D) shown by dashed boxes. (C, D) Enlarged images showing a top-down (XY) view of the ridges with the larger actin bundles regularly spaced in between the splayed ridges. Coloured lines indicate the location of the orthogonal views; with the yellow line corresponding to the XZ orthogonal view (below) and the cyan line corresponding to the YZ orthogonal view (right). White pixels indicate colocalization between the actin and the chitin in the orthogonal views. Red line through the ridges in the XZ orthogonal view corresponds to the measured intensities (E,F) of actin and chitin by position (μm). Scale bar lengths: (A,B) $5 \mu\text{m}$; (C,D,) $2 \mu\text{m}$.

In order to emphasise the importance of our study, we have included more imaging data on the *Parides arcas* scales, with cross-sections for a number of scales and intensity profiles of actin and chitin. These show that there is a very clear link between the location and formation of chitin, based on the actin precursor template. To be clear this association is seen in the ridge, the bottom right image (F) shows an orthogonal view with clear indication of the merged fluorescence channels. This new figure (R2) has been included as a new figure in the manuscript, figure 7. Along with the following text

This is particularly evident in the scales of *P. arcas* (Fig 6A-C; Fig 7, S9); whose ridge layers exhibit noticeably greater width and a more splayed-out configuration compared to *H. sara* and *M. helenor*, where the ridge layers form tightly packed multilayer reflectors. In *P. arcas* these additional actin filaments are positioned directly within the forming ridges, dorsal to the larger actin bundles and exhibit a flared pattern that mirrors the exact arrangement of the cuticle ridge layers (Fig 6Ai; Fig 7; S9). The co-imaging of chitin and actin together with orthogonal views of the ridges (Fig 7C-D; S9) confirms that these actin filaments co-localize with the cuticle ridge layers.

Furthermore, this is made easier to discern by spatially quantifying the intensity of the actin and chitin in the ridges (Fig 7E,F). The actin signal is between the peaks of chitin signal that correspond to the sides of the ridge, highlighting the presence of actin directly within the ridge structure itself. Together, these observations provide additional support for the direct templating role F-actin plays in the creation of cuticle ridge layers, which aligns with our explanation for *H. sara* (Fig 5 A, Ai).

To further highlight the universality of this we have included a further reference that helps to validate our proposed mechanism. As this same actin templating process is seen in diatoms, a wholly unrelated class of much simpler biological organisms, but importantly using the very same patterning process. Showing that this approach could be a conserved patterning route across the tree of life (Tesson et al.).

The following has been added

This same patterning approach, involving templating by the actin cytoskeleton, is also used by diatoms, a wholly unrelated class of much simpler biological organisms (54). Therefore, this process could be a conserved (or convergent) patterning route for complex nano- and microstructures across the tree of life.

Reference (54) Extensive and Intimate Association of the Cytoskeleton with Forming Silica in Diatoms: Control over Patterning on the Meso- and Micro-Scale. Tesson B, Hildebrand M (2010) Extensive and Intimate Association of the Cytoskeleton with Forming Silica in Diatoms: Control over Patterning on the Meso- and Micro-Scale. PLOS ONE 5(12): e14300.

Figure R3. The disruption of the ridges, from figure 7 in the manuscript parts M and I. Where M is the control and I is a Cyto D treated individual.

Importantly, we have also shown strong perturbation of the final chitin ridge structure with chemical disruption of actin, by using the chemical Cytochalasin D at ~ 50% development. This manipulation has severe consequences for the registry and structural colour of the fully developed ridges. This is key to showing the direct role that actin plays in the formation of structural colour in *Heliconius*. Firstly, in the SEM images we have presented in Fig. R3, we see ridge disruption and waviness due to actin perturbation.

Figure R4. A comparison of the treatment versus control of the wings using the chemical disruption with Cytochalasin D. In this protocol only the wing highlighted in A, with the red dashed line is treated, the other dorsal forewing and the hindwings have developed and show - an iridescent colour similar to the control in panel C.

The test has been revised

From the individual spectra plots (S10), most cyto-D treated individuals exhibited a completely flat reflectance spectrum with no change in angular intensity (i.e., no iridescence) and so loss of iridescent structural colour (S10). Any remaining reflected colour is bluer than the untreated specimens (bluey green). As such, the perturbation of actin using cyto-D has most likely prevented the multilayer ridges from forming their optimal spacing.

Outside the question of novelty, the manuscript is logically arranged and adds to the growing amount of literature that investigates the in vivo formation of butterfly wing scales, aimed at a specialised audience interested in this question. Some revisions are suggested by this reviewer:
- Introduction: the introduction can be put on a wider context, e.g. by including more review literature on structural colors (thinking of e.g. the McDougal, Schroeder or Isapour reviews),

The reviews mentioned by this reviewer have been included in the revised version as references 2-4

1. McDougal, A., Miller, B., Singh, M. & Kolle, M. Biological growth and synthetic fabrication of structurally colored materials. *J. Opt.* **21**, 073001 (2019).

2. Schroeder, T. B. H., Houghtaling, J., Wilts, B. D. & Mayer, M. It's Not a Bug, It's a Feature: Functional Materials in Insects. *Adv. Mater.* **30**, e1705322 (2018).

3. Isapour, G. & Lattuada, M. Bioinspired Stimuli-Responsive Color-Changing Systems. *Adv. Mater.* **30**, e1707069 (2018).

a wider introduction of growth mechanisms (at least mention alternative growth hypotheses for lumen structures), and specify a bit more that ridge multilayers are quite common before you jump into details in lines 72.

We have amended the manuscript at line 60 to include more on how common ridge multilayers are, which seems the logical place to expand on this point, also we have highlighted the inner lumen structures and the current thinking about how these are postulated to form.

Line 60: “Ghiradella (16) postulated that developing ridges buckle due to intracellular stress, and that this is responsible for the formation of layered lamellae. These lamellae act as multilayer optical reflector structures which are widely distributed and numerous across butterfly species and this structure has the flexibility to produce colours that span the optical spectrum, from the UV through to the visible spectrum. Inner lumen structures tend to be more optically and structurally complex and there are still many open questions as to how these structures form. Several studies have suggested that these may be patterned by an internal membrane structures formed by the smooth endoplasmic reticulum (11, 17), but measurements on adults scales of *Thecla opisena* suggested that cuticle extrusion and folding must be simultaneous processes (22). However, there have yet to be any direct measurements on developing scales to confirm these hypotheses and in-situ experiments of the developing inner lumen structures are needed to confirm this definitively. In addition, recently F-actin bundles have also been implicated in the formation of elaborate honeycomb nanostructures, specific to Papilionidae (23).

For reference here is the text at Line 72 - In butterfly scales, the actin bundles may not just be limited to guiding ridge positioning but could be crucial in sculpting finer-scale aspects of scale morphology, including the photonic nanostructures. Recently F-actin bundles have also been implicated in the formation of elaborate honeycomb nanostructures, specific to Papilionidae.

- The x-ray tomography dataset seems a bit superfluous (and is not shown in Figure 2, see line 92). Why is it useful and what added data can be extracted from this that couldn't be obtained from SEM images?

The x-ray tomography data helps validate the SEM measurements by confirming that there are consistent scale wide differences between the ridge spacing for the iridescent and non-iridescent scales. It also gives a 3D dataset that others may want to use in future optical model studies or studies of the hierarchy of components (trabeculae, lower lamina) in fully developed scale cells .

- Scale volume: line 116 discusses scale spacing etc. Could this be a volume conservation effect? Why yes/no?

It is slightly unclear what the question is here. If the reviewer is asking whether the decrease in ridge space can be explained simply by a reduction in scale width: Figure R5 shows that for iridescent and non-iridescent scales, there is divergence between the number of ridges and ridge width as well as the relationship between scale width and ridge spacing, so no, the decrease in ridge spacing is not a by-product of having overall smaller scales. We also find

overall more and wider ridges in the iridescent scales compared to the black ones, so there is not conservation of volume within the ridges.

If the question is whether the increased number of ridges explains the slight decrease in width of the blue scales, through an overall conservation of the volume (or surface area) of the cell when there are more buckles on the surface; this is an interesting possibility, but we are not aware of a method that would allow us to quantify the volume contained within the ridges precisely enough to answer this (and at what developmental stage it would be appropriate to do this - almost certainly not the adult scale).

Line 116 “The decreased ridge spacing in iridescent scales can be attributed to an overall increase in ridge number, rather than a smaller scale width, with iridescent scales consistently having a greater ridge number for a given scale width (Fig 2D).”

Figure R5. The relationship between scale width and ridge number, showing that it differs depending on whether the scale is iridescent or non-iridescent.

- The differentiation of non/iridescent scales in lines 130-137 needs more work and is hard to follow.

Previous version lines 130-137

Whilst the general morphology of adult *H. sara* iridescent and non-iridescent scales is similar, morphological differences in the scale ridges are observed, with adult iridescent scales displaying slightly thicker ridges and considerably reduced ridge spacing compared to non-iridescent scales. We were not able to quantify differences in the layering of lamellae within the

ridges, which has previously been shown to be responsible for the iridescent colour (30, 31), as this was beyond the resolution of the x-ray nano-tomography and would have required TEM sections of the ridges. However, the increased density and number of ridges likely contributes to the increased reflectance of the iridescent scales (30, 36).

Revised version....

In general, the morphology of adult *H. sara* iridescent and non-iridescent scales is similar, however there are distinct differences in respect of the ridges. Iridescent scales display slightly thicker ridges and reduced ridge spacing compared to non-iridescent scales. We did not quantify differences in the layering of lamellae within the ridges, which is responsible for the iridescent colour (34, 35), as this was beyond the resolution of the x-ray nano-tomography and would have required TEM sections of the ridges. The increase in ridge density and the number of ridges contributes to the increased reflectance of the iridescent scales (34, 40).

- Chitin and cuticle: these are used synonymously here, yet there is a distinct difference in its biochemical composition and function (see Politi and others). Please rephrase and be careful throughout.

We have read one of the Politi papers highlighted by this reviewer, which states that *“The exoskeletal cuticle of arthropods, based mostly on a composite of chitin filaments and protein, is a remarkable example of this versatility [6].”*

We utilised a fluorescent label that specifically binds to chitin, this is using the chitin binding domain. As such, in our imaging work we refer to chitin which forms part of the cuticle. In a number of places we have amended cuticle to chitin and vice versa. These are listed below.

Cuticle has been changed to chitin: page 9
in the region between the chitin ridges

Cuticle has been changed to chitin ridges: page 9
between adjacent chitin ridges along...

Cuticle has been changed to chitin page 13
when the chitin ridges

- Errors: many values / times of development can only be seen as approximate as its extrapolated from the mean time to the eclosion of the butterfly. I would at least add a cautious note about the times.

The *Heliconius* and *Morpho* butterflies we studied were raised from pupae in our own growth facilities at the University of Sheffield, this was all in well regulated temperature and light controlled environments. As such the pupation and eclosion time is highly consistent, with only small differences of the order of a few hours compared to the total time of pupation 8-9 days (*Heliconius*), 14 days (*Morpho*). The *Parides arcas* is different as it was ordered from a breeder, so this an approximation, this is highlighted in the methods section, see below.

Line 429 Parides arcas pupae were purchased from Stratford-upon-Avon Butterfly Farm, United Kingdom. Recording of the exact pupation times was not possible due to direct purchase from a supplier but these were approximated based on observation of the wing cuticle levels at dissection and the average time of emergence of the remaining pupae.

- Parides work: The findings of the section starting at line 235 should be compared to ref 29, but also other Parides work, e.g. 10.1186/s12862-014-0160-9

We have amended this to reflect the comments of the reviewers and have included the references they mention.

Line 235 onwards

Actin has universal patterning mechanisms across multiple butterfly species.

To confirm the universality of actin patterning mechanisms across butterfly species, we again used confocal and TauSTED microscopy to image developing wings scales of *Morpho helenor* and *Parides arcas*. We chose to study black *P. arcas* scales as these have pronounced ridge structures. A recent study examining developing *P. arcas* green scales using fluorescence imaging saw actin reorganisation similar to our observations in *H. sara*, but around honeycomb structures, which act as optical diffusers, and sit above the gyroid structural colour elements (23). Other studies (43) have seen wide variation and diversity in the types of photonic structure exhibited in species of *Parides*, including gyroid, lumen multilayer and ridge reflectors. A possible explanation for this variability of inner lumen structures has been suggested by tuning composition, similar to synthetic block copolymer nanostructures, in which the morphology is determined by the control of the constituent block volume fractions (44).

We observed similarity in the patterning and reorganisation of the actin cytoskeleton across all species of butterfly studied (Fig 6). At 50% of development, large actin bundles are present between the forming cuticle ridges in both *P. arcas* (Fig 6A-C) and *M. helenor* (Fig 6G-I). As previously described in *H. sara* and other butterfly species, these large actin bundles play a role in specifying the location of the cuticle ridges (21, 39). During this developmental stage, we also note the presence of additional actin filaments located between the larger actin bundles in the region where cuticle ridge deposition is occurring. This is particularly evident in the scale of *P. arcas* (Fig 6A-C); whose ridge layers exhibit noticeably greater width and a more splayed-out configuration compared to *H. sara* and *M. helenor*, where the ridge layers form tightly packed multilayer reflectors. In *P. arcas* these additional actin filaments are positioned directly within the forming ridges, dorsal to the larger actin bundles and exhibit a flared pattern that mirrors the arrangement of the cuticle ridge layers (Fig 6Ai). The co-imaging of chitin and actin (Fig 6C) confirms that these actin filaments precisely co-localize with the cuticle ridge layers. This observation provides additional support for the direct templating role F-actin plays in the creation of cuticle ridge layers, which aligns with our explanation for *H. sara* (Fig 5 A, Ai). At ~60-65% of development in both *P. arcas* (Fig6 D-F), and *M. helenor* (Fig6 J-L), the large bundles of actin between the cuticle ridges have dissociated (Fig6 D-F) and a highly branched network of actin filaments is visible, as was seen for *H. sara* wing scales at the equivalent development stage (Fig 4G-I). In *P. arcas*, we observe a comparable branched actin network to

that in *H. sara* scales at a corresponding developmental stage (Fig 6Di, F; Fig 4Gi, I). This network features an internal arrangement of branched actin filaments extending outward toward the cell edge from singular points located several microns within the scale. In *M. helenor*, the branched network is most discernible in the region between the cuticle ridges (Fig 6Ji, L), with F-actin regularly interspersed in a perpendicular arrangement between adjacent cuticle ridges along the length of the scale, similar to what is seen at a comparable stage in *H. sara*. Overall, our observations in three phylogenetically distinct butterfly species and different scale types, confirms the universality of a complex and highly dynamic network of actin cytoskeleton in developing scales. In all three species, the actin cytoskeleton displays similarity in its patterning and rearrangement, prefiguring diverse scale ultrastructures throughout scale cell formation.

- The cyto-D experiments are important and take up most of the discussion of the work. While interesting, the loss of structural color needs to be explained more. Some reflectance is still there and the reflectance curves certainly don't point to "a total loss of structural color" (l. 285).

AP It is important to highlight that not all injections of cyto D were successful, as such it is clear that individuals 4B and 6B show strong reflectance and iridescence (based on the peak reflectance changing intensity as a function of reflectance measurement angle).

Figure. R6 Reflectance with the two individuals removed (4B and 6B).

We are grateful for the reviewer in this comment, as they are correct to say that there is still some evidence of structural colour. However, what is clear is that it is not by-in-large iridescent (having little to no angular colour change) and also the reflected colour is bluer than the untreated specimens (bluey green). As such, the perturbation of actin has prevented the multilayer ridges from forming their optimal spacing. We have revised this in our resubmission.

Line 283: “From the individual spectra plots (S9), most cyto-D treated individuals exhibited a completely flat reflectance spectrum with no change in angular intensity (i.e., no iridescence) and so loss of iridescent structural colour loss (S9).”

Changed to

Line 283: From the individual spectra plots (S10), most cyto-D treated individuals exhibited a completely flat reflectance spectrum with no change in angular intensity (i.e., no iridescence) and so loss of iridescent structural colour (S10). Any remaining reflected colour is bluer than the untreated specimens (bluey green). As such, the perturbation of actin using cyto-D has most likely prevented the multilayer ridges from forming their optimal spacing.

- Materials: add pupation time for *P. arcas* (not given).

These have been added in addition to the *Morpho* pupation time.

Wing scale development occurs during the pupal stage (51). At the desired stage, wings were dissected from pupae in phosphate buffered saline (PBS) and immediately fixed for 15 minutes in 4 % paraformaldehyde in PBS, at room temperature. Developmental stages of pupae were recorded as a percentage of pupal development, with *H. sara* and *M. helenor* taking 8 and 14 days from pupal case formation to eclosion at 25 °C, respectively. *Parides arcas* pupae were

purchased from Stratford-upon-Avon Butterfly Farm, United Kingdom. Recording of the exact pupation times was not possible due to direct purchase from a supplier but these were approximated based on observation of the wing cuticle levels at dissection and the average time of emergence of the remaining pupae. The pupation time for Parides arcas can range from 14 - 21 days (56).

Reviewer #2 (Remarks to the Author):

This paper delves into describing the contribution of the Actin cytoskeleton into the building of intricate butterfly wing scale ultrastructures. Using groundbreaking phenotyping methods to quantifying morphology in the sub-micron range, and super-resolution microscopy (the TauSTED documentation of chitin/actin co-stains are outstanding), the authors radically raise the standard of how chitin modifications and epithelial structures are elaborated. One of the important insights that is brought up is that actin network organization continues to scaffold scale morphology throughout its development, while lower resolution data previously implied depolymerization in the late stages. The authors also complement the H. sara data (documenting a proximo-distal shift of melanic scales, from iridescent to non-iridescent), with two comparative models (Parides and Morpho), which primes interesting future studies (similarly to the recent Seah and Saranathan paper). It is also important to highlight that the article generates a lot of quantifications and measurements, and made a real effort to analyze these with proper statistics.

While technical in its Results, the article is superbly written and illustrated, with a real effort to bring clarity where it can. I thus expect it to be interest to a general audience of cell biologists interested in the quirks of actin dynamics as well as other researchers generally curious about intracellular morphogenesis. I highly recommend this article for publication, and think it does not require a significant revision.

We thank the reviewer for their positive and enthusiastic evaluation of our work and its importance.

Minor comments :

For reproducibility, the methods should further expand the description of the staining protocols, for example detailing the detergent concentration, the incubation times, and the order in which these incubations are made.

These details are available in the supplementary methods and we have included a sentence to highlight wash times and detergent concentration.

Immunofluorescence

Fixed wings (4% PFA in PBS (phosphate buffered saline)) were washed (5 minutes – 1 hour) several times in PBSTx (0.5-1% Triton-X 100). Detergent concentration and wash times were varied depending on developmental stage; with tissues with more cuticle present having longer wash times and increased detergent. For microtubule staining, samples were then blocked in 5 % Goat serum in PBSTx, rocking at room temperature for two hours. Primary antibody labelling was conducted using a mouse Anti- α -Tubulin antibody (T6199, Sigma) at 1:1000 in PBSTx, overnight at 4 °C. Secondary antibody incubation used Cy3 AffiniPure Donkey Anti-Mouse (Jackson ImmunoResearch) at 1:300 and samples were incubated at room temperature for 2-3 hours. For staining of the actin cytoskeleton / membrane / cuticle wings were left overnight at 4 °C in Phalloidin (Alexa Fluor 555/ ATTO-647; Invitrogen) or SiR-actin (Spirochrome) at 1:200 in PBS and/or Wheat Germ Agglutinin (WGA) (Alexa Fluor 647 conjugate; Invitrogen) at 1:300 in PBS. For the STED microscopy dye concentrations were increased by 2-5 fold and WGA was replaced by a Chitin Binding Domain (TMR) at a 1:75 – 1:100 dilution in PBS (New England Biolabs; special request). Finally, DAPI (1 μ g/mL) was used for counterstaining. Wings were mounted onto slides with Fluoroshield (Merck) for confocal microscopy and Mowiol or Prolong Diamond Antifade (Invitrogen) for STED microscopy and a coverslip applied. Left hindwings were used as controls, following the above protocol but omitting the fluorescent dye and/or primary antibody.

In the discussion of Fig.6, I think the article was not particularly explicit about whether the actin data supports a role in building the Braggs reflector structure itself. Is this because of a staging issue? Since Morpho iridescent scales consist of extraordinary lamellar stacks, I would have expected periodic actin branches pretemplating these extensions. I suspect this will be the subject of further studies, including perhaps in UV-iridescent pierid butterflies described in the Ghiradella foundational papers with TEM, or in other butterflies convergently evolved for ridge stacking (suggested reference : <https://doi.org/10.1242/jeb.245940>)

The Morpho we studied sadly does not have the extraordinary lamellar stacks. This is the subject of ongoing work. See the earlier figure R2, which clearly shows the actin / chitin co-localisation in the developing ridge scale for a *Parides arcas*. This new figure (figure 7) has been included in the revised manuscript.

Reviewer #3 (Remarks to the Author):

In this manuscript, the authors investigated the structural features of butterfly wing scales with a variety of techniques (e.g., SEM, X-rays, TauSTED) and propose that actin plays multiple roles in structural color formation in butterfly wing scales. The experiments were very well executed and the micrographs are very impressive. However, it remains only in a descriptive work, and the biology of the relationship between actin and butterfly wing scales remains unclear. I feel that the present form of this paper is not suitable for publication in Nature Communications.

Major concerns

1. Structural color is a well-known research field in nanophotonics and several important results have been published. Structural color is mainly dominated by various features of the cell of nanostructures, such as size, shape, categories of materials and so on. In a way, the scales of butterfly plays the similar role as the periodic cells of nanostructure. As literatures (23,25,28) reported that the actin cytoskeleton is important in controlling the number and shape of ridges in bristles. Therefore, I don't doubt the conclusion that actin cytoskeleton plays key roles in structural color formation in butterfly wing scales. In my opinion, the novelty of this work is not sufficient to attract wide attention. Moreover, the authors just propose one potential prospect: providing likely ridge patterning mechanisms. I would encourage the authors to introduce more great and further potential applications for being of interesting to the related scientific communities.

We have further highlighted the importance of this as a general patterning route and think this also addresses the comment from this reviewer. Understanding the generality of this route and understanding the way in which this templating approach could be harnessed would give the kind of technological reach and allow potential optical and sensing applications using programmed design of hierarchical nano and micro structures.

While actin has previously been shown to play a role in controlling the number and patterning of ridges, it has not previously been directly demonstrated to be involved in photonic nanostructure formation, after the ridges are present and patterned. While this had been proposed previously (Ghiradella), it was suggested to act through force-generation by the actin cytoskeleton, while we show for the first time that actin may directly templated the nanostructures.

2. This article gave a lot of descriptive work, but I think the evidence to support the important role of actin in the formation of scale-like patterns is insufficient. The only functional experiment was that cytochalasin D could disrupt the pattern of scales, but it also did not rule out the possibility that its effects were due to cell death or other indirect causes.

The scales are fully formed scales, so no, they have not died. The protocol we used was taken from the pioneering work of Dinwiddie et al. This approach is a tried and tested method for destabilising actin. It destabilised the scale formation, at the time point we used cyto d it strongly affects the scale ridges, in their alignment and registry, and also their structural colour. The control injections and the fact that only 1 wing is injected in the developing pupae allows us to compare them.

I think it is necessary to observe the influence of cytochalasin D on actin and the pattern of chitin distribution in a short period of time. Authors should also try to stabilize actin with jasplakinolide and observe its effects.

The use of Jasplakinolide would in effect give a strong perturbation of the actin globule filament equilibrium and cause a change in the actin cytoskeleton but in the opposite direction, providing hyper stable actin filaments that then aggregate. So we are not sure why this would be superior to cyto d to examine the effects on disrupting actin.

The kind of temporal experiment to look at the dynamics of actin disruption proposed by the reviewer could be interesting follow-up experiments, but this was beyond the scope of the current study.

Minor concerns

1. I think there is difference of Young's modulus between adult and developing *H. sara*. I wonder that if Young's modulus of butterfly wing scales would affect the actin cytoskeleton? Why?

We agree that there is probably a big difference between the Young's modulus for developing scale cells and mature adult ones, given the difference in hydration and the switch from actin to chitin as the scale reaches maturity and dries.

2. Lines 478 mentions that “we constructed a linear mixed effect model for each response variable...”, the actin cytoskeleton as a function of scale area, scale length, scale width et al. should be a nonlinear effect. As a result, the data interpretation of this simulated model is not absolutely accurate and sometimes wrong. I recommend the authors to add several description and explanations on this point.

The linear mixed effect model was used to analyse the fully developed scales with chitin present (n=800) for scale width, scale length and scale area, ridge spacing and ridge width separately, with respect to scale type (ground/cover and blue/black). These models were not used to test the relationships between these parameters, which we agree may not all be expected to scale linearly with respect to each other. It was not possible to measure these parameters for the developing scale cells for the actin cytoskeleton, due to the limited field of view in confocal.

Line 478 All statistical analyses were performed in R (Version 3.5.2) (50). For SEM analyses of adult iridescent and non-iridescent *H. sara* scales, we constructed a linear mixed effect model for each response variable (scale area, scale length, scale width, ridge spacing, ridge width) using the lme4 package (51). Prior to fitting the mixed effect model for ridge width, we averaged individual ridge measurements per scale. For models of ridge spacing, scale area and ridge width we included ‘individual’ as an intercept only random effect and for the model of ridge spacing, we included an interaction term between scale type and region. For scale length and scale width we fitted a random slope mixed model, allowing a different response to wing region for each individual. We used likelihood ratio tests between models with the Chi squared distribution to assess statistical significance of sequentially dropped terms. For pairwise comparisons, Tukey multiple comparison tests were performed using the emmeans package in R (52). For analyses of the ridge spacing between the proximal and distal scales of *H. e. demophon*, we firstly averaged measurements for each region per individual. Given the lower sample size we performed a paired t-test.

Changes

In the methodology we have added in a citation for the Huygens software and further details on the creation of the orthogonal views.

TauSTED microscopy

Super resolution imaging was performed on a Leica TCS SP8 STED microscope with Falcon module. Post processing of images was performed in Huygens Professional [Version 23.10](http://svi.nl) (Scientific Volume Imaging, The Netherlands, <http://svi.nl>). Images were firstly stabilized for lateral drift and the signal-to-noise ratio (SNR) was estimated for each image with acuity mode on. A conservative deconvolution strategy was selected for the estimation of initial values and the recommended CMLE (Classic Maximum Likelihood Estimation) was used as the deconvolution algorithm. Background was automatically estimated using a 0.7 μm area. A widefield search mode was used to identify the most in focus plane for background estimation. The PSF (point spread function) was estimated using a conservative optimization approach. The estimated parameters of background and PSF were then used for the final deconvolution, which was performed using an optimized iteration mode with a default quality of 0.01 for the maximum number of iterations. Orthogonal views were generated using the Ortho Slicer 2D visualisation in Huygens, using a maximum intensity projection (MIP) and channel contrast set to 'compress'.

Additional Figure 7 for parides and corresponding text changes

At 50% of development, large actin bundles are present between the forming cuticle ridges in both *P. arcas* (Fig 6A-C) and *M. helenor* (Fig 6G-I). As previously described in *H. sara* and other butterfly species, these large actin bundles play a role in specifying the location of the cuticle ridges (21, 39). During this developmental stage, we also note the presence of additional actin

filaments located between the larger actin bundles in the region where cuticle ridge deposition is occurring. This is particularly evident in the scale of *P. arcas* (Fig 6A-C; Fig 7, S9); whose ridge layers exhibit noticeably greater width and a more splayed-out configuration compared to *H. sara* and *M. helenor*, where the ridge layers form tightly packed multilayer reflectors. In *P. arcas* these additional actin filaments are positioned directly within the forming ridges, dorsal to the larger actin bundles and exhibit a flared pattern that mirrors the arrangement of the cuticle ridge layers (Fig 6Ai; Fig 7; S9). The co-imaging of chitin and actin (Fig 7C-D; S9) confirms that these actin filaments co-localize with the cuticle ridge layers. Furthermore, intensity profiles of actin and chitin (Fig 7E, F) across the ridges confirm the presence of actin directly within the ridge structure. These observations provide additional support for the direct templating role F-actin plays in the creation of cuticle ridge layers, which aligns with our explanation for *H. sara* (Fig 5 A, Ai).

Reviewer #2:

Remarks to the Author:

This is a strong revision and I recommend it for publication

Reviewer #3:

Remarks to the Author:

For my major concerns, the authors did not give a compelling response, either objecting or simply explaining. I still maintain that the novelty of this work is not sufficient. The work mainly showed a lot of descriptive work. They did not delve into the mechanism. The only functional experiment using cytochalasin D is insufficient. Thus, I still feel that the present form of this paper is not suitable for publication in Nature Communications.

Reviewer #3 (Remarks to the Author):

For my major concerns, the authors did not give a compelling responses, either objecting or simply explaining. I still maintain that the novelty of this work is not sufficient. The work mainly showed a lot of descriptive work. They did not delve into the mechanism. The only functional experiment using cytochalasin D is insufficient. Thus, I still feel that the present form of this paper is not suitable for publication in Nature Communications.

Major concerns from reviewer 3 (from the first round)

1. Structural color is a well-known research field in nanophotonics and several important results have been published. Structural color is mainly dominated by various features of the cell of nanostructures, such as size, shape, categories of materials and so on. In a way, the scales of butterfly plays the similar role as the periodic cells of nanostructure. As literatures (23,25,28) reported that the actin cytoskeleton is important in controlling the number and shape of ridges in bristles. Therefore, I don't doubt the conclusion that actin cytoskeleton plays key roles in structural color formation in butterfly wing scales. In my opinion, the novelty of this work is not sufficient to attract wide attention. Moreover, the authors just propose one potential prospect: providing likely ridge patterning mechanisms. I would encourage the authors to introduce more great and further potential applications for being of interesting to the related scientific communities.

We have further highlighted the importance of this as a general patterning route and think this also addresses the comment from this reviewer. Understanding the generality of this route and understanding the way in which this templating approach could be harnessed would give the kind of technological reach and allow potential optical and sensing applications using programmed design of hierarchical nano and micro structures.

While actin has previously been shown to play a role in controlling the number and patterning of ridges, it has not previously been directly demonstrated to be involved in photonic nanostructure formation, after the ridges are present and patterned. While this had been proposed previously (Ghiradella), it was suggested to act through force-generation by the actin cytoskeleton, while we show for the first time that actin may directly template the nanostructures.

In responding again to this reviewer's first concern we note that they agree that *"the actin cytoskeleton plays key roles in structural color formation in butterfly wing scales"*. Our claim is that actin is responsible for sculpting and controlling the placement and registry of the developing components before they are fully formed and before the scales have the appropriate level of mechanical integrity to maintain their final shape and spacing, which is what's needed to make highly reflective and coherent nanostructures. This now comes down to an editorial decision about the significance of this work and its importance.

We also do not see the need *"to introduce more great and further potential applications for being of interesting to the related scientific communities"*. It is well known how important such photonic structures are and so being able to understand their formation may help to realise ways in which we and others can begin to directly pattern and template their formation, in new areas of synthetic photonic structure formation, but this is certainly more relevant after the work is published.

2. This article gave a lot of descriptive work, but I think the evidence to support the important role of actin in the formation of scale-like patterns is insufficient. The only functional experiment was that cytochalasin D could disrupt the pattern of scales, but it also did not rule out the possibility that its effects were due to cell death or other indirect causes.

The scales are fully formed scales, so no, they have not died. The protocol we used was taken from the pioneering work of Dinwiddie et al. This approach is a tried and tested method for destabilising actin. It destabilised the scale formation, at the time point we used cyto d as it strongly affects the scale ridges, in their alignment and registry, and also their structural colour. The control injections and the fact that only 1 wing is injected in the developing pupae allows us to compare them.

The point at which we perform the functional experiment with cyto d is at 50% of scale development, importantly this is prior to chitin being deposited (~60%-70% scale development). If the cells had died then we would not expect the deposition of chitin to take place and go on to form the cuticle ridge; the SEM images in figure A1 confirm that scale cell formation has progressed after the cyto d injections have taken place.

Text from our manuscript

The pupae were injected at 50% development, after ridge spacing is set but before ridge ultrastructures form and during incipient chitin deposition, to assess the effects of actin disruption specifically on structural colour production.

Figure A1. SEM images taken from figure 8 in Lloyd et al, clearly showing the presence of chitin ridges after treatment with cytochalasin d.

If the scale cells were dead after injection they would not have reached their final scale size, we already stated that they had formed fully. In order to show how the butterfly has developed after treatment with cytochalasin d, the image below shows the specimen 7E (fig. A2) and allows

clear and direct visual comparison of the treated right dorsal forewing with the left dorsal forewing. The corresponding reflectance of the treated dorsal forewing is shown in fig. A3.

Figure A2. Photograph of a cytochalasin treated *H. sara*. The cyto d treatment has been injected into the right dorsal forewing.

Figure A3. The measured reflectance spectra for all cyto d treated individuals (and controls), this shows the reflectance for the individual (7E) shown in figure A2.

The use of cytochalasin D to inhibit the development of actin polymerization has been known about for over 35 years, this has mainly been used in the manipulation and treatment of developing fly bristles, see (Cooper et al 1987).

Turner and Adler state “For example the treatment of developing microvilli with cytochalasin D, a known inhibitor of actin polymerization (Cooper, 1987) results in the formation of branched microvilli (Burgess and Grey, 1974)”.

Figure A4. data taken from Fig. 6 in Turner and Adler (1998). This shows how cytochalasin d treatment affects bristle formation in *Drosophila*. (A) Wild-type bristles. (B) Application of 1 mM cytochalasin D caused a stunted or bent morphology in the bristles. (C) Occasionally branched bristles could be found after cytochalasin D treatment (arrow points to branch).

Recent work examining both scale and bristle development in mosquitoes (*Aedes aegypti*) by Djokic et al (2020), used a very similar method to assess function, cyto d followed by SEM.

The use of cycto d to perturb actin formation is also well known in the butterfly scale development literature, as actin has a prominent role in sculpting the larger scale wide structures and layout of the scale, this was shown in the work of Dinwiddie et al (2014). This work looked at butterfly scales that were not ridge reflectors. We saw no need to collect timepoints earlier than 50 % scale development in our study as this would result in complete scale cell disruption, as has already been seen by Dinwiddie in 2014, see figure A5.

Figure A5. Fluorescence microscopy data taken from Dinwiddie et al 2014 (figure 9). Treatment with cytochalasin D at 24 hours after pupation, the actin filaments within the cells of the wing epithelium are disrupted and do not elongate.

I think it is necessary to observe the influence of cytochalasin D on actin and the pattern of chitin distribution in a short period of time.

We are not entirely sure what is meant by this comment, it can either mean that we examine the scales at earlier developmental time points or that we undertake a time series of scale development (after actin perturbation by cyto d), by imaging them after pupation. We would argue that this study has already been done by Dinwiddie et al and to have a very high standard and so this would not be novel. In the developmental stages that are comparable to our scales

these were clearly alive, with membranes intact. They studied cyto d injections at 24hrs, 48hrs, 54hrs, 66hrs, and 96hrs. They also examined different stages and time points in the development of the scales after injection, this is shown in figure A5 above.

A temporal experiment to look at the dynamics of actin disruption and the effects on cuticle disruption after cyto-D injection but before adult emergence at super-resolution H. sara (with ridge reflectors) could be interesting follow-up experiments, but this was beyond the scope of the current study, and would take at least 6 months of further experimental work (for which we lack the time and resources).

Authors should also try to stabilize actin with jasplakinolide and observe its effects.

The use of Jasplakinolide would in effect give a strong perturbation of the actin globule filament equilibrium and cause a change in the actin cytoskeleton but in the opposite direction, providing hyper stable actin filaments that then aggregate. So we are not sure why this would be superior to cyto d to examine the effects on disrupting actin.

Nevertheless, we have undertaken Jasplakinolide treatment of the developing scales and saw no effect based on the sample size studied, (11 treated, 10 controls). We could not see local perturbation or whole scale changes to scale development. It may be that the higher log P of this molecule means it does not have suitable solubility (logp 5.6) to enter the scale cells and may become partitioned in lipid membranes, and so not disrupt the actin, whereas cytochalasin d (logp 2.7) does not do this to the same extent. Significant further experimental work would be needed to optimise these experiments and determine why we see no effect.

We have written an experimental method and discussion of this experiment which we are happy to append to the supporting information in our paper should the reviewers think that it is appropriate.

References

- C.M. Turner, P.N. Adler. Distinct roles for the actin and microtubule cytoskeletons in the morphogenesis of epidermal hairs during wing development in *Drosophila* Mech. Dev., 70, 181-192, (1998).
- J. A. Cooper, J.A.. Effects of cytochalasin and phalloidin on actin. J. Cell Biol. 105, 1473–1478 (1987).
- A. Dinwiddie, R. Null, M. Pizzano, L. Chuong, A. L. Krup, H. E. Tan, N. H. Patel, Dynamics of F-actin prefigure the structure of butterfly wing scales, Dev. Biology, 392, 404-418, (2014).
- Djokic S, Bakhrat A, Tsurim I, Urakova N, Rasgon JL, Abdu U. Actin bundles play a different role in shaping scales compared to bristles in the mosquito *Aedes aegypti*. Sci Rep. 10;10(1):14885. (2020).

Report on Jasplakinolide treatment of developing wings

Jasplakinolide was injected into the developing pupae following the same procedure used for the injection of cytochalasin D. Pupae were injected in their proximal forewing at approximately 50% development (day 3.5) with either 5 µl of Graces insect medium (control) or 5 µl of 5 µM Jasplakinolide in Graces insect medium (treated). This concentration is in line with previous

experiments using Jasplakinolide to perturb bristle development in *Drosophila* (1–3) and well above the minimum concentration known to produce discernible effects to the actin within cultured cells (50-100 nM) (4). After injection with Jasplakinolide the pupae were allowed to continue their development and eclosion. The pupae emerged at day 7 and 8 and after several hours were humanely culled.

10/12 control pupae emerged and 12/13 treated pupae emerged with no effect of Jasplakinolide on pupal mortality. Visual inspection of the wings revealed no obvious local or widespread effect on the wing colouration of the treated samples compared to the control. In both control and treated wings, it was obvious that the blue iridescence still remained in the proximal forewing. This contrasts to our experiments with cytochalasin D which changed the colour hue of the entire proximal forewing (i.e. the whole iridescent region appeared darker purple – black in individuals which were treated with cytochalasin D). While there were noticeable deformities present on some of the treated wings, these are almost certainly due to the mechanical damage produced from the insertion of the needle through the pupal case and into the wing tissue. This is most damaging when the needle tip punctures a wing vein, causing considerable damage to scales within the vicinity of the vein. However, this is also observed in the control samples and therefore such scale effects can be attributed entirely to the mechanical needle insertion procedure rather than a consequence of the drug treatment. Given the lack of visual difference between the control samples and the Jasplakinolide samples, it can be concluded that there has been minimal to no effect of the drug on the optical nanostructures within the wing scales.

This lack of effect may be due to several reasons, which are not necessarily mutually exclusive. Firstly, the concentration may not have been large enough to produce an effect. While the concentration selected and used was in line with previous studies, these were conducted in cultured cells and tissues and not whole organisms. Secondly, it may be that the higher log P of the Jasplakinolide molecule means it does not have sufficient solubility (logp 5.6) to effectively disrupt the scale cells as it may become partitioned and trapped in lipid membranes, such as the epicuticle layer surrounding the scales, whereas cytochalasin D (logp 2.7) does not do this to the same extent. Finally, the particular conformation and properties of actin within the scale cells may be inherently resistant to the biochemical mode of action of the drug. Bubb et al., (1994) concluded that Jasplakinolide has a much greater effect on Mg^{2+} -actin than on Ca^{2+} -actin (5). Furthermore, Rotsch and Radmacher (2000) found that cytochalasins and related compounds were capable of disassembling stress fibers but that Jasplakinolide did not disassemble stress fibers (6). At the scale developmental stage injected (50%) the actin was fully assembled into large actin bundles (stress fibers) but this was immediately prior to the formation of branched actin networks, it could be that the Jaspakinolide molecules bonded with the larger stress fibers with no resultant biomechanical consequence.

Figure 1. Jasplakinolide (5 μ M) injection into pupae at 50% development had no observable effect on the blue iridescent colour of the proximal forewing in *H. sara* (blue box) compared to controls which were injected with Graces insect medium (orange box).

1. L. G. Tilney, P. S. Connelly, L. Ruggiero, K. A. Vranich, G. M. Guild, D. Stocks, Actin Filament Turnover Regulated by Cross-linking Accounts for the Size , Shape , Location , and Number of Actin Bundles in *Drosophila* Bristles. **14**, 3953–3966 (2003).
2. L. G. Tilney, P. S. Connelly, G. M. Guild, Microvilli appear to represent the first step in actin bundle formation in *Drosophila* bristles (2004), doi:10.1242/jcs.01215.
3. G. M. Guild, P. S. Connelly, K. A. Vranich, M. K. Shaw, L. G. Tilney, Actin filament turnover removes bundles from *Drosophila* bristle cells. *J. Cell Sci.* **115**, 641–653 (2002).

4. M. R. Bubb, I. Spector, B. B. Beyer, K. M. Fosen, Effects of jasplakinolide on the kinetics of actin polymerization. An explanation for certain *in vivo* observations. *J. Biol. Chem.* **275**, 5163–5170 (2000).
5. M. R. Bubb, A. M. J. Senderowicz, E. A. Sausville, K. L. K. Duncan, E. D. Korn, Jasplakinolide, a cytotoxic natural product, induces actin polymerization and competitively inhibits the binding of phalloidin to F-actin. *J. Biol. Chem.* **269**, 14869–14871 (1994).
6. C. Rotsch, M. Radmacher, Drug-induced changes of cytoskeletal structure and mechanics in fibroblasts: An atomic force microscopy study. *Biophys. J.* **78**, 520–535 (2000).

Reviewers' Comments:

Reviewer #3:

Remarks to the Author:

No more comments from my side.